# RNNs of RNNs:
# Recursive Construction of Stable Assemblies of Recurrent Neural Networks

**Leo Kozachkov**[1,*], **Michaela Ennis**[2,*], and **Jean-Jacques Slotine**[1,3,4]

[1]Department of Brain and Cognitive Sciences, Massachusetts Institute of Technology
[2]Division of Medical Sciences, Harvard University
[3]Department of Mechanical Engineering, Massachusetts Institute of Technology
[4]Google AI
[*]Equal contribution
{leokoz8,mennis,jjs}@mit.edu

## Abstract

Recurrent neural networks (RNNs) are widely used throughout neuroscience as models of local neural activity. Many properties of single RNNs are well characterized theoretically, but experimental neuroscience has moved in the direction of studying multiple interacting *areas*, and RNN theory needs to be likewise extended. We take a constructive approach towards this problem, leveraging tools from nonlinear control theory and machine learning to characterize when combinations of stable RNNs will themselves be stable. Importantly, we derive conditions which allow for massive feedback connections between interacting RNNs. We parameterize these conditions for easy optimization using gradient-based techniques, and show that stability-constrained 'networks of networks' can perform well on challenging sequential-processing benchmark tasks. Altogether, our results provide a principled approach towards understanding distributed, modular function in the brain.

## 1 Introduction

The combination and reuse of primitive "modules" has enabled a great deal of progress in computer science, engineering, and biology. Modularity is particularly apparent in the structure of the brain, as different parts are specialized for different functions [Kandel et al., 2000]. Accordingly, most experimental studies throughout the history of neuroscience have focused on a single brain area in association with a single behavior [Abbott and Svoboda, 2020]. Similarly, RNN models of brain function have mostly been limited to a single RNN modeling a single area. However, neuroscience is entering an age where recording from many different brain areas simultaneously during complex behaviors is possible. As experimental neuroscience has shifted towards multi-area recordings, computational techniques for analyzing, modeling, and interpreting these multi-area recordings have blossomed [Michaels et al., 2020, Mashour et al., 2020, Abbott and Svoboda, 2020, Perich et al., 2021, Semedo et al., 2019, Yang and Molano-Mazón, 2021, Machado et al., 2022]. Despite this, RNN theory has lagged behind.

The theoretical question of RNN stability is crucial for understanding information propagation and manipulation [Vogt et al., 2020, Engelken et al., 2020, Kozachkov et al., 2022a]. The conditions under which single, autonomous RNNs are chaotic or stable are well-studied, in particular when the RNN weights are randomly chosen and the number of neurons tends to infinity [Sompolinsky et al., 1988, Engelken et al., 2020]. However, there is very little work addressing the theoretical

question of stability in 'networks of networks'. Two facts make this question challenging. Firstly, connecting two stable systems does not, in general, lead to a stable overall system. This is true even for linear systems. Secondly, there is a massive amount of feedback between brain areas, so one cannot reasonably assume near-decomposability [Simon, 1962, Abbott and Svoboda, 2020].

Given that the brain seems to dynamically reorganize and adapt interareal connectivity to meet task demands and environmental constraints [Miller and Cohen, 2001, Sych et al., 2022], this question of how stability is maintained is of the utmost importance. Here we take a bottom-up approach, more specifically asking "what stability properties of the individual modules lend themselves to rapid reorganization?"

## 1.1 Contraction Analysis

We focus on a special type of stability, known as contractive stability [Lohmiller and Slotine, 1998]. Loosely, a contracting system is a dynamical system that forgets its initial conditions exponentially quickly. Contractive stability is a strong form of dynamical stability which implies many other forms of stability, such as certain types of input-to-state stability [Sontag, 2010]. See Section A.2 for a mathematical primer on contraction analysis.

Contraction analysis has found wide application in nonlinear control theory [Manchester and Slotine, 2017], synchronization [Pham and Slotine, 2007], and robotics [Chung and Slotine, 2009], but has only recently begun to find application in neuroscience and machine learning [Boffi et al., 2020, Wensing and Slotine, 2020, Kozachkov et al., 2020, Revay and Manchester, 2020, Jafarpour et al., 2021, Kozachkov et al., 2022a, Centorrino et al., 2022, Burghi et al., 2022, Kozachkov et al., 2022b]. Contraction analysis is useful for neuroscience because it is directly applicable to systems with external inputs. It also allows for modular stability-preserving *combination* properties to be derived (Figure 1). The resulting contracting combinations can involve individual systems with different dynamics, as long as those dynamics are contracting [Slotine and Lohmiller, 2001, Slotine, 2003].

Moreover, modular stability and specifically contractive stability have relevance to evolutionary biology [Simon, 1962, Slotine and Lohmiller, 2001]. In particular, it is thought that the majority of traits that have developed over the last 400+ million years are the result of evolutionary forces acting on regulatory elements that combine core components, rather than mutations in the core components themselves. This mechanism of action makes meaningful variation in population phenotypes much more feasible to achieve, and is appropriately titled "facilitated variation" [Gerhart and Kirschner, 2007]. In addition to the biological evidence for facilitated variation, computational models have demonstrated that this approach produces populations which are better able to generalize to new environments [Parter et al., 2008], an ability that will be critical to further develop in deep learning systems. However, the tractability of these evolutionary processes hinges on some mechanism for ensuring stability of combinations. Because contraction analysis tools allow complicated contracting systems to be built up recursively from simpler elements, this form of stability would be well suited for biological systems [Slotine and Liu, 2012]. Our work with the Sparse Combo Net in Section 4 has direct parallels to facilitated variation, in that we train this combination network architecture *only* through training connections between contracting subnetworks.

Ultimately, our contributions are three-fold:

- A novel parameterization for feedback combination of contracting RNNs that enables direct optimization using standard deep learning libraries.
- Novel contraction conditions for continuous-time nonlinear RNNs, to use in conjunction with the combination condition. We also identify flaws in stability proofs from prior literature.
- Experiments demonstrating that our 'network of networks' sets a new state of the art for stability-constrained RNNs on benchmark sequential processing tasks.

## 2 Network of Networks Model

In this paper we analyze rate-based neural networks. Unlike spiking neural networks, these models are continuous and smooth. We consider the following RNN introduced by Wilson and Cowan [1972], which may be viewed as an approximation to a more biophysically-detailed spiking network:

$$\tau \dot{\mathbf{x}} = -\mathbf{x} + \mathbf{W}\phi(\mathbf{x}) + \mathbf{u}(t) \tag{1}$$

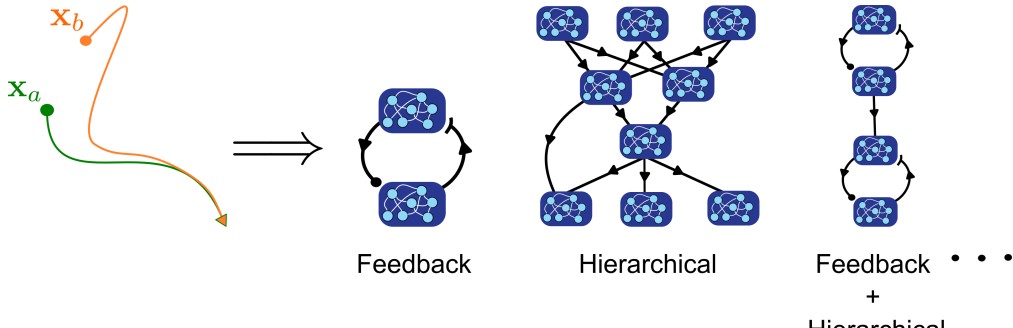

Figure 1: Contractive stability implies a modularity principle. Because contraction analysis tools allow complicated contracting systems to be built up recursively from simpler elements, this form of stability is well suited for understanding biological systems. Contracting combinations can be made between systems with very different dynamics, as long as those dynamics are contracting.

Here $\tau > 0$ is the time-constant of the network [Dayan and Abbott, 2005], and the vector $\mathbf{x} \in \mathbb{R}^n$ contains the voltages of all $n$ neurons in the network. The voltages are converted into firing-rates through a static nonlinearity $\phi$. We only consider monotonic activation functions with bounded slope: in other words, $0 \leq \phi' \leq g$ (unless otherwise noted). We do not restrain the sign of the nonlinearity. Common example nonlinearities $\phi(x)$ that satisfy these constraints are hyperbolic tangent and ReLU. The matrix $\mathbf{W} \in \mathbb{R}^{n \times n}$ contains the synaptic weights of the RNN. It is this matrix that ultimately determines the stability of (1), and will be a main target for our analysis. Finally, $\mathbf{u}(t)$ is the potentially time-varying external input into the network, capturing both explicit input into the RNN from the external world, as well as unmodeled dynamics from other brain areas.

Note that (1) is equivalent to another commonly used class of RNNs where the term $\mathbf{Wx} + \mathbf{u}$ appears inside the nonlinearity. See Section A.1 or [Miller and Fumarola, 2012] for details. Our mathematical results apply equally well to both types of RNNs.

In order to extend (1) into a model of multiple interacting neural populations, we introduce the index $i$, which runs from 1 to $p$, where $p$ is the total number of RNNs in the collection of RNNs. For now we will assume linear interactions between RNNs, because this is the simplest case. The linearity assumption can also be motivated by the fact that RNNs have been found to be well-approximated by linear systems in many neuroscience contexts [Sussillo and Barak, 2013, Langdon and Engel, 2022]. This leads to the following equation for the 'network of networks':

$$\tau \dot{\mathbf{x}}_i = -\mathbf{x}_i + \mathbf{W}_i \phi(\mathbf{x}_i) + \sum_{j=1}^{p} \mathbf{L}_{ij} \mathbf{x}_j + \mathbf{u}_i(t) \quad \forall i = 1 \cdots p \tag{2}$$

where the matrix $\mathbf{L}_{ij}$ captures the interaction between RNN $i$ and RNN $j$. If RNN $i$ has $N_i$ neurons and RNN $j$ has $N_j$ neurons, then $\mathbf{L}_{ij}$ is an $N_i \times N_j$ matrix.

We can now formalize the question posed in the introduction. Namely, "what types of connections between stable RNNs automatically preserve stability?" becomes "what restrictions on $\mathbf{L}_{ij}$ must be met in order to ensure overall stability of the network of networks?". We will now derive two combination 'primitives', negative feedback and hierarchical, which allow for recursive connection of contracting modules.

## 2.1 Generalized Negative Feedback Between RNNs Preserves Stability

We set aside for a moment the problem of determining when (1) is contracting. For now, assume that we have a collection of $p$ contracting RNNs interacting through equation (2). Recall that contraction is defined with respect to a *metric*, a way of measuring distances between trajectories in state space. Thus, the $i$th RNN is contracting with respect to some metric $\mathbf{M}_i$. We assume for simplicity that this metric is constant, which means that $\mathbf{M}_i$ is simply a symmetric, positive definite matrix. In the case where every RNN receives feedback from every other, we can preserve stability by ensuring these connections are negative feedback. In the simplest case where all RNN modules are contracting in

the identity metric, the negative feedback may be written as:

$$\mathbf{L}_{ij} = -\mathbf{L}_{ji}^T$$

This is a well known result from the contraction analysis literature [Slotine, 2003]. Our first novel contribution is a generalization and parameterization of this negative feedback which allows for direct optimization using gradient-based techniques. In particular, if each $\mathbf{L}_{ij}$ is parameterized as follows:

$$\mathbf{L}_{ij} = \mathbf{B}_{ij} - \mathbf{M}_i^{-1}\mathbf{B}_{ji}^T\mathbf{M}_j \qquad \forall i, j \tag{3}$$

for arbitrary matrix $\mathbf{B}_{ij}$, then the overall network of networks retains the assumed contraction properties of the RNN subnetworks. We provide a detailed proof in Section C.1, but the basic idea relies on ensuring skew-symmetry of $\mathbf{L}_{ij}$ *in the appropriate metric*. This can be achieved via the constraint:

$$\mathbf{M}_i\mathbf{L}_{ij} = -\mathbf{L}_{ji}^T\mathbf{M}_j$$

Plugging (3) into the above expression verifies that it is indeed satisfied. Because contraction analysis relies on analyzing the symmetric part of Jacobian matrices, the skew-symmetry of $\mathbf{L}_{ij}$ 'cancels out' when computing the symmetric part, and leaves the stability of the subnetwork RNNs untouched. In the remainder of the paper, in the experimental sections, we will use feedback combinations constrained by (3). However, it is possible to significantly generalize this condition (see C.1 for details).

**Recursive Properties of Contracting Combinations** The feedback combination (3), taken together with hierarchical combinations, may be used as combination primitives for recursively constructing complicated networks of networks while automatically maintaining stability. The recursion comes from the fact that once a modular system is shown to be contracting it may be treated as a single contracting system, which may in turn be combined with other contracting systems, *ad infinitum*. Note that while the feedback (3) requires linear interareal connections, hierarchical interareal connections may be nonlinear [Lohmiller and Slotine, 1998].

## 3 Various Ways to Achieve Local Contraction

In this section we return to the question of achieving contraction in the subnetwork RNNs. Recall that we wish to find restrictions on $\mathbf{W}_i$ such that the $i$th subnetwork RNN, described by (1), is contracting. Here we derive five such novel conditions (see Section C for detailed proofs). As we will discuss, not all contraction conditions are equally useful - for example some conditions are easier to optimize or have higher model capacity than others. In this section we also point out some flaws in existing stability proofs in the RNN literature, and suggest some pathways towards correcting them.

**Theorem 1** (Absolute Value Restricted Weights)**.** *Let* $|\mathbf{W}|$ *denote the matrix formed by taking the element-wise absolute value of* $\mathbf{W}$*. If there exists a positive, diagonal* $\mathbf{P}$ *such that:*

$$\mathbf{P}(g|\mathbf{W}| - \mathbf{I}) + (g|\mathbf{W}| - \mathbf{I})^T\mathbf{P} \prec 0$$

*with* $g$ *being the maximum slope of* $\phi$*, then* (1) *is contracting in metric* $\mathbf{P}$*. If* $W_{ii} \leq 0$*, then* $|W|_{ii}$ *may be set to zero to reduce conservatism.*

It is easy to find matrices that satisfy Theorem 1, and given a matrix the condition is as easy to check as linear stability is. Moreover, the condition guarantees we can obtain a metric that the system is known to contract in (see Section 4.1 for details). It is less straightforward to enforce this condition during training, however we found that subnetworks constrained by Theorem 1 can achieve high performance in practice by simply fixing $\mathbf{W}$ and only optimizing the connections *between* subnetworks (Section 4.2). As there are fewer parameters to optimize, this training technique is faster.

**Theorem 2** (Symmetric Weights)**.** *If* $\mathbf{W} = \mathbf{W}^T$ *and* $g\mathbf{W} \prec \mathbf{I}$*, and* $\phi' > 0$*, then (1) is contracting.*

It has been known since the early 1990s that if (1) is autonomous (i.e the input $\mathbf{u}$ is constant) and has symmetric weights with eigenvalues less than $1/g$, then there exists a unique fixed point that the network converges to from any initial condition [Matsuoka, 1992]. Theorem 2 generalizes this statement to say that if (1) has symmetric weights with eigenvalues less than $1/g$, it is contracting. This includes previous results as a special case, because an *autonomous* contracting system has a unique fixed point which the network converges to from any initial condition.

**Theorem 3** (Product of Diagonal and Orthogonal Weights)**.** *If there exists positive diagonal matrices* $\mathbf{P}_1$ *and* $\mathbf{P}_2$*, as well as* $\mathbf{Q} = \mathbf{Q}^T \succ 0$ *such that*

$$\mathbf{W} = -\mathbf{P}_1\mathbf{Q}\mathbf{P}_2$$

*then (1) is contracting in metric* $\mathbf{M} = (\mathbf{P}_1\mathbf{Q}\mathbf{P}_1)^{-1}$*.*

In contrast to the first two contraction conditions, Theorem 3 is very easy to optimize. To meet the constraint that the $\mathbf{P}$ matrices are positive, one can parameterize their diagonal elements as $P_{ii} = d_i^2 + \epsilon$, for some small positive constant $\epsilon$, and optimize $d_i$ directly. To meet the positive definiteness constraint on $\mathbf{Q}$, one may parameterize it as $\mathbf{Q} = \mathbf{E}^T\mathbf{E} + \epsilon\mathbf{I}$ and optimize $\mathbf{E}$ directly.

**Theorem 4** (Triangular Weights)**.** *If* $g\mathbf{W} - \mathbf{I}$ *is triangular and Hurwitz, then (1) is contracting in a diagonal metric.*

Theorem 4 follows from the fact that a hierarchy of contracting systems is also contracting.

**Theorem 5** (Singular Value Restricted Weights)**.** *If there exists a positive diagonal matrix* $\mathbf{P}$ *such that:*

$$g^2\mathbf{W}^T\mathbf{P}\mathbf{W} - \mathbf{P} \prec 0$$

*then (1) is contracting in metric* $\mathbf{P}$*.*

In the case of discrete-time RNNs, this contraction condition has been proved by many different authors in many different settings. When $\mathbf{P} = \mathbf{I}$, it is known as the echo-state condition for discrete-time RNNs [Jaeger, 2001]. This was then generalized to diagonal $\mathbf{P}$ by Buehner and Young [2006]. More recently, the original echo-state condition was rediscovered by Miller and Hardt [2018] in the machine learning literature. Following this rediscovery, the condition was generalized to $\mathbf{P} \neq \mathbf{I}$ by Revay and Manchester [2020]. Here we show that it applies to continuous-time RNNs as well.

## 3.1 What do the Jacobian Eigenvalues Tell Us?

Several recent papers in ML, e.g [Haber and Ruthotto, 2017, Chang et al., 2019], claim that a sufficient condition for stability of the nonlinear system:

$$\dot{\mathbf{x}} = \mathbf{f}(\mathbf{x}, t)$$

is that the associated Jacobian matrix $\mathbf{J}(\mathbf{x}, t) = \frac{\partial \mathbf{f}}{\partial \mathbf{x}}$ has eigenvalues whose real parts are strictly negative, i.e:

$$\max_i \text{Re}(\lambda_i(\mathbf{J}(\mathbf{x}, t)) \leq -\alpha$$

with $\alpha > 0$. However, this claim is generally false - see Section 4.4.2 in [Slotine and Li, 1991].

In the *specific* case of the RNN (1), it appears that the eigenvalues of the symmetric part of $\mathbf{W}$ do provide information on global stability in a number of applications. For example, in [Matsuoka, 1992] it was shown that if $\mathbf{W}_s = \frac{1}{2}(\mathbf{W} + \mathbf{W}^T)$ has all its eigenvalues less than unity, and $\mathbf{u}$ is constant, then (1) has a unique, globally asymptotically stable fixed point. This condition also implies that the real parts of the eigenvalues of the Jacobian are uniformly negative. Moreover, in [Chang et al., 2019] it was shown that setting the symmetric part of $\mathbf{W}_s = \frac{1}{2}(\mathbf{W} + \mathbf{W}^T)$ almost equal to zero (yet slightly negative) led to rotational, yet stable dynamics in practice. This leads us to the following theorem, which shows that if the slopes of the activation functions change sufficiently slowly as a function of time, then the condition in [Matsuoka, 1992] in fact implies global contraction of (1).

**Theorem 6.** *Let* $\mathbf{D}$ *be a positive, diagonal matrix with* $D_{ii} = \frac{d\phi_i}{dx_i}$*, and let* $\mathbf{P}$ *be an arbitrary, positive diagonal matrix. If:*

$$(g\mathbf{W} - \mathbf{I})\mathbf{P} + \mathbf{P}(g\mathbf{W}^T - \mathbf{I}) \preceq -c\mathbf{P} \quad \text{and} \quad \dot{\mathbf{D}} - cg^{-1}\mathbf{D} \preceq -\beta\mathbf{D}$$

*for* $c, \beta > 0$*, then (1) is contracting in metric* $\mathbf{D}$ *with rate* $\beta$*.*

We stress however, that it is an open question whether or not diagonal stability of $\mathbf{W}$ implies that (1) is contracting. It has been conjectured that diagonal stability of $g\mathbf{W} - \mathbf{I}$ is a sufficient condition for global contraction of (1) [Revay et al., 2020], however this has been difficult to prove. To better characterize this conjecture, we present Theorem 7, which shows by way of counterexample that diagonal stability of $g\mathbf{W} - \mathbf{I}$ does not imply global contraction in a *constant* metric for (1).

**Theorem 7.** *Satisfaction of the condition* $g\mathbf{W}_{sym} - \mathbf{I} \prec 0$ *is* ***not*** *sufficient to show global contraction of the general nonlinear RNN (1) in any* ***constant*** *metric. High levels of antisymmetry in* $\mathbf{W}$ *can make it impossible to find such a metric, which we demonstrate via a $2 \times 2$ counterexample of the following form, with $c \geq 2$ when $g = 1$:* $\mathbf{W} = \begin{bmatrix} 0 & -c \\ c & 0 \end{bmatrix}$

## 4 Stability-Constrained Network of Networks Perform Well on Benchmarks

A natural concern is that stability of an RNN may come at the cost of its expressivity, which is particularly relevant for integrating information over long timescales. To investigate whether this might be an issue for our model, we trained a stability-constrained network-of-networks on three benchmark sequential image classification tasks: sequential MNIST, permuted seqMNIST, and sequential CIFAR10. These tasks are often used to measure information processing ability over long sequences [Le et al., 2015]. Images are presented pixel-by-pixel, and the network makes a prediction at the end of the sequence. In permuted seqMNIST, pixels are input in a fixed but random order.

All of our experiments were done on networks governed by (2). The nonlinear subnetwork RNNs were connected to each other via linear all-to-all negative feedback, given by (3). For all subnetworks we use the ReLU activation function. To enforce contraction of each individual subnetwork, we focused on two stability constraints from our theoretical results: Theorems 1 and 5. In the case of Theorem 1, we did not train the individual subnetworks' weight matrices, but only trained the connections *between* subnetworks (Figure 2B). For Theorem 5, we trained all parameters of the model (Figure 2C).

We refer to networks with subnetworks constrained by Theorem 1 as 'Sparse Combo Nets' and to networks with subnetworks constrained by Theorem 5 as 'SVD Combo Nets'. Throughout the experimental results we use the notation '$p \times n$ network' - such a network consists of $p$ distinct subnetwork RNNs, with each such subnetwork RNN containing $n$ units.

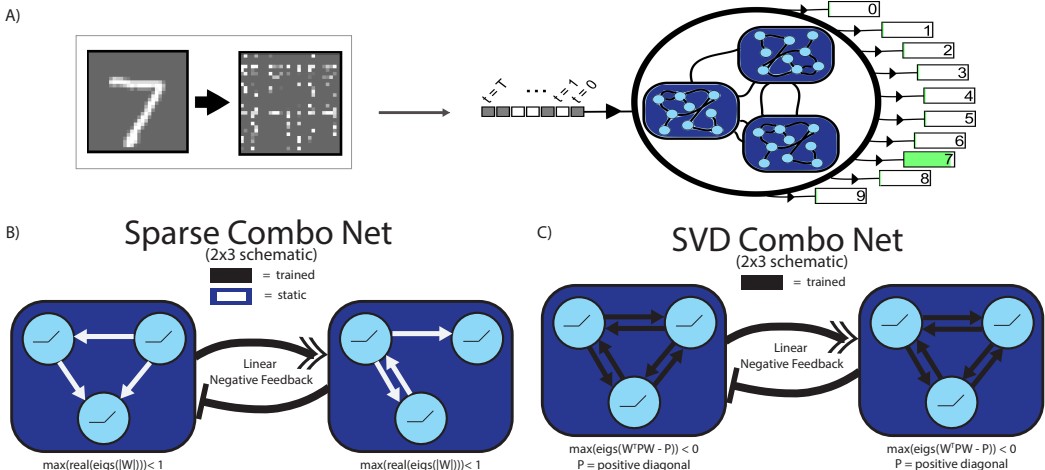

Figure 2: Summary of task structure and network architectures. Images from MNIST (or CIFAR10) were flattened into an array of pixels and fed sequentially into the modular 'network of networks', with classification based on the output at the last time-step. For MNIST, each image was also permuted in a fixed manner (A). The subnetwork 'modules' of our architecture were constrained to meet either Theorem 1 via sparse initialization (B) or Theorem 5 via direct parameterization (C). Linear negative feedback connections were trained between the subnetworks according to (3).

### 4.1 Network Initialization and Training

For the Sparse Combo Net we were not able to find a parameterization to continuously update the internal RNN weights during training in a way that preserved contraction. However, it is easy to randomly generate matrices with a particular likelihood of meeting the Theorem 1 condition by selecting an appropriate sparsity level and limit on entry magnitude. Sparsity in particular is of interest

due to its relevance in neurobiology and machine learning, so it is convenient that the condition makes it easy to verify stability of many different sparse RNNs. As $g = 1$ for ReLU activation, we check potential subnetwork matrices $\mathbf{W}$ by simply verifying linear stability of $|\mathbf{W}| - \mathbf{I}$.

Because every RNN meeting the condition has a corresponding well-defined stable LTI system contracting in the same metric, it is also easy to find a metric to use in our training algorithm: solving for $\mathbf{M}$ in $-\mathbf{I} = \mathbf{MA} + \mathbf{A}^T\mathbf{M}$ will produce a valid metric for any stable LTI system $\mathbf{A}$ [Slotine and Li, 1991]. We utilize the fact that Hurwitz Metzler matrices are diagonally stable to improve efficiency of computing $\mathbf{M}$ (as well as in our proof of Theorem 1).

We therefore randomly generated fixed subnetworks satisfying Theorem 1 and trained only the linear connections between them (Figure 3), as well as the linear input and output layers. More information on network initialization, hyperparameter tuning, and training algorithm is provided in Section D.

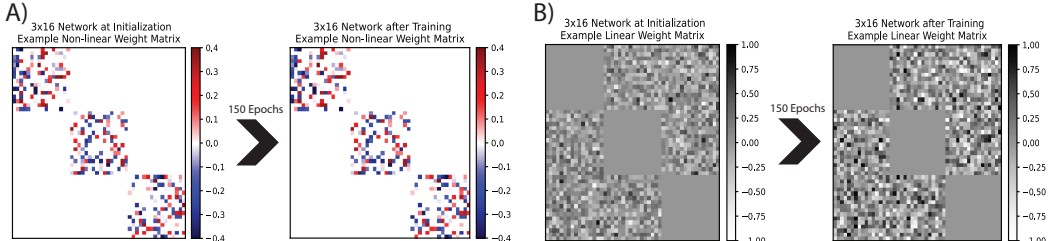

Figure 3: Example 3x16 Sparse Combo Net. Nonlinear intra-subnetwork weights are initialized using a set sparsity, and do not change in training (A). Linear inter-subnetwork connections are constrained to be antisymmetric with respect to the overall network metric, and are updated in training (B).

For the SVD Combo Net on the other hand, we ensured contraction of each subnetwork RNN by direct parameterization (described in Section E), thus allowing all weights to be trained.

## 4.2 Results

The Sparse Combo Net architecture achieved the highest overall performance on both permuted seqMNIST and seqCIFAR10, with 96.94% and 65.72% best test accuracies respectively - thereby setting a new SOTA for stable RNNs (Table 1). Furthermore, we were able to reproduce SOTA scores over several repetitions, including 10 trials of seqCIFAR10. Along with repeatability of results, we also show that the contraction constraint on the connections between subnetworks ($\mathbf{L}$ in (3)) is important for performance, particularly in the Sparse Combo Net (Section 4.2.3).

Additionally, we profile how various architecture settings impact performance of our networks. In both networks, we found that increasing the total number of neurons improved task performance, but with diminishing returns (Section 4.2.1). We also found that the sparsity of the hidden-to-hidden weights in Sparse Combo Net had a large impact on the final network performance (Section 4.2.2).

### 4.2.1 Experiments with Network Size

Understanding the effect of size on network performance is important to practical application of these architectures. For both Sparse Combo Net and SVD Combo Net, increasing the number of subnetworks while holding other settings constant (including fixing the size of each subnetwork at 32 units) was able to increase network test accuracy on permuted seqMNIST to a point (Figure 4).

The greatest performance jump happened when increasing from one module (37.1% Sparse Combo Net, 61.8% SVD Combo Net) to two modules (89.1% Sparse Combo Net, 92.9% SVD Combo Net). After that the performance increased steadily with number of modules until saturating at $\sim 97\%$ for Sparse Combo Net and $\sim 95\%$ for SVD Combo Net.

As the internal subnetwork weights are not trained in Sparse Combo Net, it is unsurprising that its performance was substantially worse at the smallest sizes. However Sparse Combo Net surpassed SVD Combo Net by the $12 \times 32$ network size, which contains a modest 384 total units. Due to the better performance of the Sparse Combo Net, we focused additional analyses there. Note also that the SVD Combo Net never reached 55% test accuracy for CIFAR10 in our early experiments.

| Name | Stable RNN? | Params | sMNIST Repeats Mean (n) [Min] | psMNIST Repeats Mean (n) [Min] | sCIFAR10 Repeats Mean (n) [Min] | Seq MNIST Best | PerSeq MNIST Best | Seq CIFAR Best |
|---|---|---|---|---|---|---|---|---|
| LSTM [Chang et al., 2019] | | 68K | — | — | — | 97.3% | 92.7% | 59.7% |
| Transformer [Trinh et al., 2018] | | 0.5M | — | — | — | 98.9% | 97.9% | 62.2% |
| Antisymmetric [Chang et al., 2019] | ? | 36K | — | — | — | 98% | 95.8% | 58.7% |
| Sparse Combo Net | ✓ | 130K | — | **96.85%** (4) [**96.65%**] | 64.72% (10) [63.73%] | 99.04% | **96.94%** | **65.72%** |
| Lipschitz [Erichson et al., 2021] | ✓ | 134K | 99.2% (10) [99.0%] | 95.9% (10) [95.6%] | — | **99.4%** | 96.3% | 64.2% |
| CKConv [Romero et al., 2021] | | 1M | — | — | — | 99.32% | 98.54% | 63.74% |
| S4 [Gu et al., 2022] | | 7.9M | — | — | — | **99.63%** | **98.7%** | **91.13%** |
| Trellis [Bai et al., 2019] | | 8M | — | — | — | 99.2% | 98.13% | 73.42% |

Table 1: Published benchmarks for sequential MNIST, permuted MNIST, and sequential CIFAR10 best test accuracy. Architectures are grouped into three categories: baselines, best performing RNNs with claimed stability guarantee*, and networks achieving overall SOTA. Within each grouping, networks are ordered by number of trainable parameters (for CIFAR10 if it differed across tasks). Our network is highlighted. Where possible, we include information on repeatability.
*For more on stability guarantees in machine learning, see Section 3.1

We then evaluated task performance as the *modularity* of a Sparse Combo Net (fixed to have 352 total units) was varied. We observed an inverse U shape (Figure S1B), with poor performance of a $1 \times 352$ net and an $88 \times 4$ net, and best performance from a $44 \times 8$ net. However, this experiment compared similar sparsity levels, while in practice we can achieve better performance with larger subnetworks by leveraging sparsity in a way not possible for smaller ones.

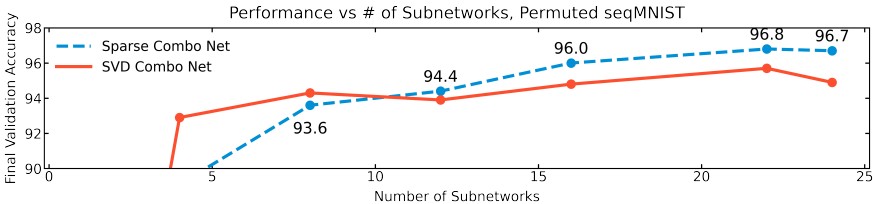

Figure 4: Permuted seqMNIST performance plotted against the number of subnetworks. Each subnetwork has 32 neurons. Results are shown for both Sparse Combo Net and SVD Combo Net.

### 4.2.2 Experiments with Sparsity

Because of the link between sparsity and stability as well as the biological relevance of sparsity, we explored in detail how subnetwork sparsity affects the performance of Sparse Combo Net. We ran a number of experiments on the permuted seqMNIST task, varying sparsity level while holding network size and other hyperparameters constant. Here we use "$n\%$ sparsity level" to refer to a network with subnetworks that have just $n\%$ of their weights non-zero.

We observed a large ($> 5$ percentage point) performance boost when switching from a $26.5\%$ sparsity level to a $10\%$ sparsity level in the $11 \times 32$ Sparse Combo Net (Figure 5), and subsequently decided to test significantly sparser subnetworks in a $16 \times 32$ Sparse Combo Net. We trained networks with sparsity levels of $5\%$, $3.3\%$, and $1\%$, as well as $10\%$ for baseline comparison (Figure S2A). A $3.3\%$ sparsity level produced the best results, leading to our SOTA performance for stable networks on both permuted seqMNIST and seqCIFAR10. With a component RNN size of just 32 units, this sparsity level is small, containing only one or two directional connections per neuron on average (Figure S7).

As sparsity had such a positive effect on task performance, we did additional analyses to better understand why. We found that decreasing the magnitude of non-zero elements while holding sparsity level constant decreased task performance (Figure S2B), suggesting that the effect is driven in part by the fact that sparsity enables higher magnitude non-zero elements while still maintaining stability.

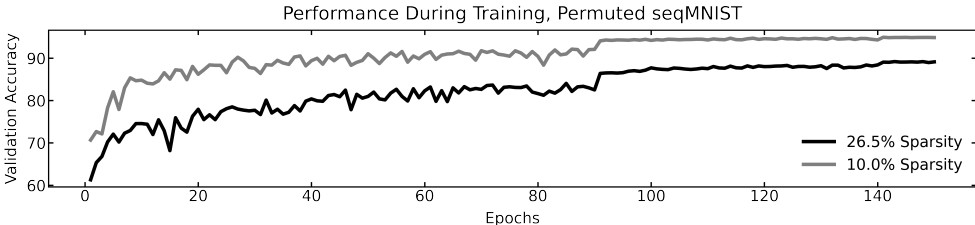

Figure 5: Permuted seqMNIST performance over the course of training for two $11 \times 32$ Sparse Combo Nets with different sparsity levels.

The use of sparsity in subnetworks to improve performance suggests another interesting direction that could enable better scalability of total network size - enforcing sparsity in the linear feedback weight matrix ($\mathbf{L}$). We performed pilot testing of this idea, which showed promise in mitigating the saturation effect seen in Figure 4. Those results are detailed in Section D.2.4 (Table S1).

### 4.2.3   Repeatability and Controls

Sparse Combo Net does not have the connections within its subnetworks trained, so network performance could be particularly susceptible to random initialization. Thus we ran repeatability studies on permuted sequential MNIST and sequential CIFAR10 using our best network settings ($16 \times 32$ with subnetwork sparsity level of $3.3\%$) and an extended training period. Mean performance over 4 trials of permuted seqMNIST was $96.85\%$ with $0.019$ variance, while mean performance over 10 trials of seqCIFAR10 was $64.72\%$ with $0.406$ variance. Note we also ran a number of additional experiments on size and sparsity settings, described in Section D.2.

Across the permuted seqMNIST trials, best test accuracy always fell between $96.65\%$ and $96.94\%$, a range much smaller than the differences seen with changing sparsity settings and network size. Three of the four trials showed best test accuracy $\geq 96.88\%$, despite some variability in early training performance (Figure S3). Similarly, eight of the ten seqCIFAR10 trials had test accuracy $> 64.3\%$, with all results falling between $63.73\%$ and $65.72\%$ (Figure S4). This robustly establishes a new SOTA for stable RNNs, comfortably beating the previously reported (single run) $64.2\%$ test accuracy achieved by Lipschitz RNN [Erichson et al., 2021].

As a control study, we also tested how sensitive the Sparse Combo Net was to the stabilization condition on the interconnection matrix ($\mathbf{L}$ in (3)). To do so, we initialized the individual RNN modules in a $24 \times 32$ network as before, but set $\mathbf{L} = \mathbf{B}$ and did not constrain $\mathbf{B}$ at all during training, thus no longer ensuring contraction of the overall system. This resulted in $47.0\%$ test accuracy on the permuted seqMNIST task, a stark decrease from the original $96.7\%$ test accuracy - thereby demonstrating the utility of the contraction condition.

## 5   Discussion

Biologists have long noted that modularity provides organisms with stability and robustness [Kitano, 2004]. The other direction – that stability and robustness provide modularity – is well known to engineers [Khalil, 2002, Slotine and Li, 1991, Slotine, 2003], but has been less appreciated in biology.

We use this principle to build and train provably stable assemblies of recurrent neural networks. Like real brains, the components of our "RNN of RNNs" can communicate with one another through a mix of hierarchical and feedback connections. In particular, we theoretically characterized conditions under which an RNN of RNNs will be stable, given that each individual RNN is stable. We also provided several novel stability conditions for single RNNs that are compatible with these stability-preserving interareal connections. Our results contribute towards understanding how the brain maintains stable and accurate function in the presence of massive interareal feedback, as well as external inputs.

The question of neural stability is one of the oldest questions in computational neuroscience. Indeed, cyberneticists were concerned with this question before the term 'computational neuroscience' existed [Wiener, 1948, Ashby, 1952]. Stability is a central component in several influential neuroscience theories [Hopfield, 1982, Seung, 1996, Murphy and Miller, 2009], perhaps the most well-known being that memories are stored as stable point attractors [Hopfield, 1982]. Our work shows that stability continues to be a useful concept for computational neuroscience as the field transitions from focusing on single brain areas to many interacting brain areas.

While primarily motivated by neuroscience, our approach is also relevant for machine learning. Deep learning models can be as inscrutable as they are powerful. This opacity limits conceptual progress and may be dangerous in safety-critical applications like autonomous driving or human-centered robotics. Given that stability is a fundamental property of dynamical systems – and is intimately linked to concepts of control, generalization, efficiency, and robustness – the ability to guarantee stability of a recurrent model will be important for ensuring deep networks behave as we expect them to [Richards et al., 2018, Choromanski et al., 2020, Revay et al., 2021, Rodriguez et al., 2022].

In the case of RNNs, one difficulty is that providing a certificate of *stability* is often impossible or computationally impractical. However, the stability conditions we derive here allow for recursive construction of complicated RNNs while automatically preserving stability. By parameterizing our conditions for easy optimization using gradient-based techniques, we successfully trained our architecture on challenging sequential processing benchmarks. The high test accuracy our networks achieved with a small number of trainable parameters demonstrates that stability does not necessarily come at the cost of expressivity. Thus, our results likewise contribute towards understanding stability certification of RNNs.

In future work, we will explore how our contraction-constrained RNNs of RNNs perform on a variety of neuroscience tasks, in particular tasks with multimodal structure [Yang et al., 2019]. Our approach is particulary compatible with "global workspace" models, in which different networks communicate via a shared latent space [Newell and Simon, 1972, Baars, 1993, Dehaene et al., 1998, VanRullen and Kanai, 2021, Goyal et al., 2021]. One desiderata for these future models is that they learn representations which are formally similar to those observed in the brain [Yamins et al., 2014, Schrimpf et al., 2020, Williams et al., 2021], in complement with the structural similarities already shared. Moreover, a "network of networks" approach will be especially relevant to challenging multimodal machine learning problems, such as the simultaneous processing of audio and video. Therefore the advancement of neuroscience theory and machine learning remain hand-in-hand for our next lines of questioning. Indeed, combinations of trained networks have already seen groundbreaking success in DeepMind's AlphaGo [Silver et al., 2016].

As well as the many potential experimental applications, there are numerous theoretical future directions suggested by our work. Networks with more biologically-plausible weight update rules, such as models discussed in [Kozachkov et al., 2020], would be a fruitful neuroscience context in which to explore our conditions. One promising avenue of study there is to examine input-dependent stability of the learning process. In the context of machine learning, our stability conditions could be applied to the end-to-end training of multidimensional recurrent neural networks [Graves et al., 2007], which have clear structural parallels to our RNNs of RNNs but lack known stability guarantees.

In sum, recursively building network combinations in an effective and stable fashion while also allowing for continual refinement of the individual networks, as nature does for biological networks, will require new analysis tools. Here we have taken a concrete step towards the development of such tools, not only through our theoretical results, but also through their application to create stable combination network architectures that perform well in practice on benchmark tasks.

## Acknowledgments and Disclosure of Funding

This work benefited from stimulating discussions with Michael Happ, Quang-Cuong Pham, and members of the Fiete lab at MIT.

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
