# Appendix

# A  Extended Background

## A.1  Two Different RNNs

Note that in neuroscience, the variable $\mathbf{x}$ in equation (1) is typically thought of as a vector of neural membrane potentials. It was shown in [Miller and Fumarola, 2012] that the RNN (1) is equivalent via an affine transformation to another commonly used RNN model,

$$\tau \dot{\mathbf{y}} = -\mathbf{y} + \phi(\mathbf{W}\mathbf{y} + \mathbf{b}(t)) \tag{4}$$

where the variable $\mathbf{y}$ is interpreted as a vector of firing rates, rather than membrane potentials. The two models are related by the transformation $\mathbf{x} = \mathbf{W}\mathbf{y} + \mathbf{b}$, which yields

$$\tau \dot{\mathbf{x}} = \mathbf{W}(-\mathbf{y} + \phi(\mathbf{W}\mathbf{y} + \mathbf{b})) + \tau \dot{\mathbf{b}} = -\mathbf{x} + \mathbf{W}\phi(\mathbf{x}) + \mathbf{v}$$

where $\mathbf{v} \equiv \mathbf{b} + \tau \dot{\mathbf{b}}$. Thus $\mathbf{b}$ is a low-pass filtered version of $\mathbf{v}$ (or conversely, $\mathbf{v}$ may be viewed as a first order prediction of $\mathbf{b}$) and the contraction properties of the system are unaffected by the affine transformation. Note that the above equivalence holds even in the case where $\mathbf{W}$ is not invertible. In this case, the two models are proven to be equivalent, provided that $\mathbf{b}(0)$ and $\mathbf{y}(0)$ satisfy certain conditions–which are always possible to satisfy [Miller and Fumarola, 2012]. Therefore, any contraction condition derived for the $x$ (or $y$) system automatically implies contraction of the other system. We exploit this freedom freely throughout the paper.

## A.2  Contraction Math

It can be shown that the non-autonomous system

$$\dot{\mathbf{x}} = \mathbf{f}(\mathbf{x}, t)$$

is contracting if there exists a metric $\mathbf{M}(\mathbf{x}, t) = \mathbf{\Theta}(\mathbf{x}, t)^T \mathbf{\Theta}(\mathbf{x}, t) \succ 0$ such that uniformly

$$\dot{\mathbf{M}} + \mathbf{M}\mathbf{J} + \mathbf{J}^T \mathbf{M} \preceq -\beta \mathbf{M}$$

where $\mathbf{J} = \frac{\partial \mathbf{f}}{\partial \mathbf{x}}$ and $\beta > 0$. For more details see the main reference [Lohmiller and Slotine, 1998]. Similarly, a non-autonomous discrete-time system

$$\mathbf{x}_{t+1} = \mathbf{f}(\mathbf{x}_t, t)$$

is contracting if

$$\mathbf{J}^T \mathbf{M}_{t+1} \mathbf{J} - \mathbf{M}_t \preceq -\beta \mathbf{M}_t$$

### A.2.1  Feedback and Hierarchical Combinations

Consider two systems, independently contracting in constant metrics $\mathbf{M}_1$ and $\mathbf{M}_2$, which are combined in feedback:

$$\begin{aligned} \dot{\mathbf{x}} &= \mathbf{f}(\mathbf{x}, t) + \mathbf{B}\mathbf{y} \\ \dot{\mathbf{y}} &= \mathbf{g}(\mathbf{y}, t) + \mathbf{G}\mathbf{x} \end{aligned} \qquad \text{(Feedback Combination)}$$

If the following relationship between $\mathbf{B}, \mathbf{G}, \mathbf{M}_1$, and $\mathbf{M}_2$ is satisfied:

$$\mathbf{B} = -\mathbf{M}_1^{-1} \mathbf{G}^T \mathbf{M}_2$$

then the combined system is contracting as well. This may be seen as a special case of the feedback combination derived in [Tabareau and Slotine, 2006]. The situation is even simpler for hierarchical

combinations. Consider again two systems, independently contracting in some metrics, which are combined in hierarchy:

$$\dot{\mathbf{x}} = \mathbf{f}(\mathbf{x}, t)$$
$$\dot{\mathbf{y}} = \mathbf{g}(\mathbf{y}, t) + \mathbf{h}(\mathbf{x}, t) \qquad \text{(Hierarchical Combination)}$$

where $\mathbf{h}(\mathbf{x}, t)$ is a function with *bounded* Jacobian. Then this combined system is contracting in a diagonal metric, as shown in [Lohmiller and Slotine, 1998]. By recursion, this extends to hierarchies of arbitrary depth.

## B  Extended Discussion

Given the paucity of existing theory on modular networks, our novel stability conditions and proof of concept combination architectures are a significant step in an important new direction. The "network of networks" approach is evident in the biological brain, and has seen early practical success in applications such as AlphaGo. There is much evidence this line of questioning will be critical in the future, and our work is the first on stable *modular* networks.

Furthermore, we develop an architecture based on such combinations of "vanilla" RNNs that is both stable and achieves high performance on benchmark sequence classification tasks using few trainable parameters (small particularly for sequential CIFAR10). When considering just the facts about the network, it really has no business performing anywhere near as well as it does. Note also that without the stability condition in place, the network performance indeed drops substantially.

In order to facilitate the extension of this important line of thinking, we provide additional context on the limitations of our current approach as well as promising ideas for future directions in this section.

### B.1  Limitations

One drawback of our approach is that we parameterize each weight matrix in a special way to guarantee stability. In all cases, this requires us to parameterize matrices as the product of several other matrices. Thus, we gain stability at the cost of increasing the number of parameters, which can slow down training.

Another current drawback is that we only consider constant metrics. In theory, contraction metrics can be state-dependent as well as time-varying. Thus it is possible that we have overly restricted the space of models we consider. Similarly, negative feedback is not the only way to preserve contraction when combining two contracting systems. There are known small-gain theorems in the contraction analysis literature which accomplish the same task [Slotine, 2003]. However, parameterizing these conditions is less straightforward than parameterizing the negative-feedback condition.

A third limitation of the present work is that it does not give a recipe on how to incorporate anatomical knowledge into the building of 'RNNs of RNNs'. Our current approach is 'bottom up', in the sense that we describe complicated networks which can be built from simpler ones while ensuring stability at every level of construction. However, for building biological models of the brain it is important to incorporate known anatomical detail (i.e V4 projects to PFC, PFC projects back to V4, etc). How to do this in a way that preserves stability is an open and interesting question.

Lastly, we only tested our networks on sequential image classification benchmarks. Future work will include other benchmarks such as character or word-level language modeling. Additionally, while we conducted preliminary experiments exploring the role of scale (i.e number of subnetwork RNNs), we did not pursue this at the sizes reached by many modern deep learning applications. Thus it is currently unclear if the performance of these stability-constrained models will scale well enough with the number of subnetworks (or the number of neurons per subnetwork). Testing this correctly will require extensive experimentation with the various initialization and training settings.

### B.2  Future Directions

There are numerous future directions enabled by this work. For example, Theorem 6 suggests that a less restrictive contraction condition on W in terms of the eigenvalues of the symmetric part is

possible and desirable. Meanwhile, Theorem 7 provides important groundwork in finding such a condition, as it shows the need for a time-varying metric. Investigation of input-dependent metrics could be a fruitful next line of research, and would have far-reaching implications in disciplines such as curriculum learning.

Furthermore, the beneficial impact of sparsity on training these stable models suggests a potential avenue for additional experimental work – in particular adding a regularizing sparsity term during training. This could allow Sparse Combo Net to have its internal subnetwork weights trained without losing the stability guarantee, and conversely it could allow SVD Combo Net to reap some of the performance benefits of Sparse Combo Net without giving up the flexibility allowed by training said internal weights.

As described in the limitations above, a major experimental next step will be to test our architectures at greater scale and on more difficult tasks. Since 'network of network' approaches are becoming increasingly popular, our methodology is relevant to a variety of task types, including reinforcement learning applications. Given the biological inspiration, multi-modal learning tasks may also be of particular relevance.

# C    Proofs for Main Results

## C.1    Proof of Feedback Combination Property

Here we apply existing contraction analysis results to derive equation (3). Because (3) is a parameterization of a known contraction conditions [Slotine, 2003], we provide the following statement in the form of a corollary.

**Corollary 1** (Network of Networks)**.** *Consider a collection of $p$ subnetwork RNNs governed by (1). Assume that these RNNs each have hidden-to-hidden weight matrices $\{\mathbf{W}_1, \ldots, \mathbf{W}_p\}$ and are independently contracting in metrics $\{\mathbf{M}_1, \ldots, \mathbf{M}_p\}$. Define the block matrix $\tilde{\mathbf{W}} \equiv BlockDiag(\mathbf{W}_1, \ldots, \mathbf{W}_p)$ and $\tilde{\mathbf{M}} \equiv BlockDiag(k_1\mathbf{M}_1, \ldots, k_p\mathbf{M}_p)$ where $k_i > 0$. Also define the overall state vector $\tilde{\mathbf{x}}^T \equiv (\mathbf{x}_1^T \cdots \mathbf{x}_p^T)$, and finally $\tilde{\mathbf{u}}^T \equiv (\mathbf{u}_1^T \cdots \mathbf{u}_p^T)$. Finally, the matrix $\tilde{\mathbf{L}}$ is a block matrix constructed from the matrices $\mathbf{L}_{ij}$ by placing them at the $i, j$ block position of $\tilde{\mathbf{L}}$. Then if there exists a positive semi-definite matrix $\mathbf{Q}$ such that:*

$$\tilde{\mathbf{M}}\tilde{\mathbf{L}} + \tilde{\mathbf{L}}^T\tilde{\mathbf{M}} = -\mathbf{Q}$$

*then the following 'network of networks' is globally contracting in metric $\tilde{\mathbf{M}}$:*

$$\tau\dot{\tilde{\mathbf{x}}} = -\tilde{\mathbf{x}} + \tilde{\mathbf{W}}\phi(\tilde{\mathbf{x}}) + \tilde{\mathbf{u}} + \tilde{\mathbf{L}}\tilde{\mathbf{x}} \tag{5}$$

*Proof.* Consider the differential Lyapunov function:

$$V = \frac{1}{2}\delta\mathbf{x}^T\tilde{\mathbf{M}}\delta\mathbf{x}$$

The time-derivative of this function is:

$$\dot{V} = \delta\mathbf{x}^T\tilde{\mathbf{M}}\delta\dot{\mathbf{x}} = \delta\mathbf{x}^T(\underbrace{\tilde{\mathbf{M}}\tilde{\mathbf{J}} + \tilde{\mathbf{J}}^T\tilde{\mathbf{M}}}_{\text{Jacobian of RNNs before interconnection}} + \underbrace{\tilde{\mathbf{M}}\tilde{\mathbf{L}} + \tilde{\mathbf{L}}^T\tilde{\mathbf{M}}}_{\text{Interconnection Jacobian}})\delta\mathbf{x}$$

Since we assume that the RNNs are contracting in isolation, the first term in this sum is less that the slowest contracting rate of the individual RNNs, which we call $\lambda > 0$, scaled by the corresponding $k$. Plugging in the definition of $\mathbf{Q}$, we see the second term in the sum is negative semi-definite, by construction. The time-derivative of $V$ is therefore upper-bounded by:

$$\dot{V} \leq -2\lambda V$$

which implies that the network of networks is contracting with rate $\lambda$.    □

**Corollary 2.** *If $\tilde{\mathbf{L}}$ can be written as:*

$$\tilde{\mathbf{L}} \equiv \mathbf{B} - \tilde{\mathbf{M}}^{-1}\mathbf{B}^T\tilde{\mathbf{M}} - \tilde{\mathbf{M}}^{-1}\mathbf{C}$$

*where $\mathbf{B}$ is an arbitrary square matrix, and $\mathbf{C}$ is a matrix whose symmetric part is positive semidefinite, then the above stability criterion is satisfied*

$$\tilde{\mathbf{M}}\tilde{\mathbf{L}}+\tilde{\mathbf{L}}^T\tilde{\mathbf{M}} = \tilde{\mathbf{M}}(\mathbf{B}-\tilde{\mathbf{M}}^{-1}\mathbf{B}^T\tilde{\mathbf{M}}-\tilde{\mathbf{M}}^{-1}\mathbf{C})+(\mathbf{B}-\tilde{\mathbf{M}}^{-1}\mathbf{B}^T\tilde{\mathbf{M}}-\tilde{\mathbf{M}}^{-1}\mathbf{C})^T\tilde{\mathbf{M}} = -[\mathbf{C}+\mathbf{C}^T] = -\mathbf{Q}$$

*In this case we may identify $\mathbf{Q} = \mathbf{C} + \mathbf{C}^T$. For the experiments in the main text, we use $\mathbf{C} = \mathbf{0}$, although one could also learn $\mathbf{C}$ via a suitable parametrization.*

## C.2 Proof of Theorem 1

Our first theorem is motivated by the observation that if the y-system (described in Section A.1) is to be interpreted as a vector of firing rates, it must stay positive for all time. For a linear, time-invariant system with positive states, diagonal stability is equivalent to stability. Therefore a natural question is if diagonal stability of a linearized y-system implies anything about stability of the nonlinear system. More formally, given an excitatory neural network (i.e $\forall ij, W_{ij} \geq 0$), if the linear system

$$\dot{\mathbf{x}} = -\mathbf{x} + g\mathbf{W}\mathbf{x}$$

is stable, then there exists a positive diagonal matrix P such that:

$$\mathbf{P}(g\mathbf{W} - \mathbf{I}) + (g\mathbf{W} - \mathbf{I})^T\mathbf{P} \prec 0$$

The following theorem shows that the nonlinear system (1) is indeed contracting in metric $\mathbf{P}$, and extends this result to a more general $\mathbf{W}$ by considering only the magnitudes of the weights.

**Theorem 1.** *Let $|\mathbf{W}|$ denote the matrix formed by taking the element-wise absolute value of $\mathbf{W}$. If there exists a positive, diagonal $\mathbf{P}$ such that:*

$$\mathbf{P}(g|\mathbf{W}| - \mathbf{I}) + (g|\mathbf{W}| - \mathbf{I})^T\mathbf{P} \prec 0$$

*then (1) is contracting in metric $\mathbf{P}$. Moreover, if $W_{ii} \leq 0$, then $|W|_{ii}$ may be set to zero to reduce conservatism.*

This condition is particularly straightforward in the common special case where the network does not have any self weights, with the leak term driving stability. While it can be applied to a more general $\mathbf{W}$, the condition will of course not be met if the network was relying on highly negative values on the diagonal of $\mathbf{W}$ for linear stability. As demonstrated by counterexample in the proof of Theorem 1, it can be impossible to use the same metric $\mathbf{P}$ for the nonlinear RNN in such cases.

Theorem 1 allows many weight matrices with low magnitudes or a generally sparse structure to be verified as contracting in the nonlinear system (1), by simply checking a linear stability condition (as linear stability is equivalent to diagonal stability for Metzler matrices too [Narendra and Shorten, 2010]).

Beyond verifying contraction, Theorem 1 actually provides a metric, with little need for additional computation. Not only is it of inherent interest that the same metric can be shared across systems in this case, it is also of use in machine learning applications, where stability certificates are becoming increasingly necessary. Critically, it is feasible to enforce the condition during training via L2 regularization on $\mathbf{W}$. More generally, there are a variety of systems of interest that meet this condition but do not meet the well-known maximum singular value condition, including those with a hierarchical structure.

*Proof.* Consider the differential, quadratic Lyapunov function:

$$V = \delta\mathbf{x}^T\mathbf{P}\delta\mathbf{x}$$

where $\mathbf{P} \succ 0$ is diagonal. The time derivative of $V$ is:

$$\dot{V} = 2\delta\mathbf{x}^T\mathbf{P}\dot{\delta\mathbf{x}} = 2\delta\mathbf{x}^T\mathbf{P}\mathbf{J}\delta\mathbf{x} = -2\delta\mathbf{x}^T\mathbf{P}\delta\mathbf{x} + 2\delta\mathbf{x}^T\mathbf{P}\mathbf{W}\mathbf{D}\delta\mathbf{x}$$

where $\mathbf{D}$ is a diagonal matrix such that $\mathbf{D}_{ii} = \frac{d\phi_i}{dx} \geq 0$. We can upper bound the quadratic form on the right as follows:

$$\delta\mathbf{x}^T\mathbf{PWD}\delta\mathbf{x} = \sum_{ij} P_i W_{ij} D_j \delta x_i \delta x_j \leq$$

$$\sum_i P_i W_{ii} D_i |\delta x_i|^2 + \sum_{ij, i \neq j} P_i |W_{ij}| D_j |\delta x_i||\delta x_j| \leq g|\delta\mathbf{x}|^T\mathbf{P}|\mathbf{W}||\delta\mathbf{x}|$$

If $W_{ii} \leq 0$, the term $P_i W_{ii} D_i |\delta x_i|^2$ contributes non-positively to the overall sum, and can therefore be set to zero without disrupting the inequality. Now using the fact that $\mathbf{P}$ is positive and diagonal, and therefore $\delta\mathbf{x}^T\mathbf{P}\delta\mathbf{x} = |\delta\mathbf{x}|^T\mathbf{P}|\delta\mathbf{x}|$, we can upper bound $\dot{V}$ as:

$$\dot{V} \leq |\delta\mathbf{x}|^T(-2\mathbf{P} + \mathbf{P}|\mathbf{W}| + |\mathbf{W}|\mathbf{P})|\delta\mathbf{x}| = |\delta\mathbf{x}|^T[(\mathbf{P}(|\mathbf{W}| - \mathbf{I}) + (|\mathbf{W}|^T - \mathbf{I})\mathbf{P})]|\delta\mathbf{x}|$$

where $|W|_{ij} = |W_{ij}|$, and $|W|_{ii} = 0$ if $W_{ii} \leq 0$ and $|W|_{ii} = |W_{ii}|$ if $W_{ii} > 0$. This completes the proof.

Note that $\mathbf{W} - \mathbf{I}$ is Metzler, and therefore will be Hurwitz stable if and only if $\mathbf{P}$ exists [Narendra and Shorten, 2010].

It is also worth noting that highly negative diagonal values in $\mathbf{W}$ will prevent the same metric $\mathbf{P}$ from being used for the nonlinear system. Therefore the method used in this proof cannot feasibly be adapted to further relax the treatment of the diagonal part of $\mathbf{W}$.

The intuitive reason behind this is that in the symmetric part of the Jacobian, $\frac{\mathbf{PWD}+\mathbf{DW}^T\mathbf{P}}{2} - \mathbf{P}$, the diagonal self weights will also be scaled down by small $\mathbf{D}$, while the leak portion $-\mathbf{P}$ remains untouched by $\mathbf{D}$.

Now we actually demonstrate a counterexample, presenting a $2 \times 2$ symmetric Metzler matrix $\mathbf{W}$ that is contracting in the identity in the linear system, but cannot be contracting *in the identity* in the nonlinear system (1):

$$\mathbf{W} = \begin{bmatrix} -9 & 2.5 \\ 2.5 & 0 \end{bmatrix}$$

To see that it is not possible for the more general nonlinear system with these weights to be contracting in the identity, take $\mathbf{D} = \begin{bmatrix} 0 & 0 \\ 0 & 1 \end{bmatrix}$. Now

$$(\mathbf{WD})_{sym} - \mathbf{I} = \begin{bmatrix} -1 & 1.25 \\ 1.25 & -1 \end{bmatrix}$$

which has a positive eigenvalue of $\frac{1}{4}$.

$\square$

## C.3 Proof of Theorem 2

While regularization may push networks towards satisfying Theorem 1, strictly enforcing the condition during optimization is not straightforward. This motivates the rest of our theorems, which derive contraction results for specially structured weight matrices. Unlike Theorem 1, these results have direct parameterizations which can easily be plugged into modern optimization libraries.

**Theorem 2.** *If $\mathbf{W} = \mathbf{W}^T$ and $g\mathbf{W} \prec \mathbf{I}$, and and $\phi' > 0$ then (1) is contracting.*

When $\mathbf{W}$ is symmetric, (1) may be seen as a continuous-time Hopfield network. Continuous-time Hopfield networks with symmetric weights were recently shown to be closely related to Transformer architectures [Krotov and Hopfield, 2020, Ramsauer et al., 2020]. Specifically, the dot-product attention rule may be seen as a discretization of the continuous-time Hopfield network with softmax activation function [Krotov and Hopfield, 2020]. Our results here provide a simple sufficient (and nearly necessary, see above remark) condition for global exponential stability of a given *trajectory* for the Hopfield network. In the case where the input into the network is constant, this trajectory is a fixed point. Moreover, each trajectory associated with a unique input is guaranteed to be unique. Finally, we note that our results are flexible with respect to activation functions so long as they satisfy the slope-restriction condition. This flexibility may be useful when, for example, considering recent work showing that standard activation functions may be advantageously replaced by attention mechanisms [Dai et al., 2020].

*Proof.* We begin by writing $\mathbf{W} = \mathbf{R} - \mathbf{P}$ for some unknown $\mathbf{R} = \mathbf{R}^T$ and $\mathbf{P} = \mathbf{P}^T \succ 0$. The approach of this proof is to show by construction that the condition $g\mathbf{W} \prec \mathbf{I}$ implies the existence of an $\mathbf{R}$ and $\mathbf{P}$ such that the system is contracting in metric $\mathbf{P}$. We consider the $y$ version of the RNN, which as discussed above is equivalent to the $x$ version via an affine transformation.

The differential Lyapunov condition associated to the RNN is:

$$\delta\mathbf{x}^T[-2\mathbf{M} + \mathbf{MDW} + \mathbf{WDM} + \beta\mathbf{M}]\delta\mathbf{x} \leq 0 \tag{6}$$

Where $\mathbf{M}, \mathbf{W} \in \mathbb{R}^{n \times n}$. Let us now make the substitution $\mathbf{M} = \mathbf{P}$ and $\mathbf{W} = \mathbf{R} - \mathbf{P}$:

$$\delta\mathbf{x}^T[-2\mathbf{P} + \mathbf{PD}(\mathbf{R} - \mathbf{P}) + (\mathbf{R} - \mathbf{P})\mathbf{DP} + \beta\mathbf{P}]\delta\mathbf{x} \leq 0 \tag{7}$$

Collecting terms, we get:

$$\delta\mathbf{x}^T[-2\mathbf{P} + \mathbf{PDR} + \mathbf{RDP} - 2\mathbf{PDP} + \beta\mathbf{P}]\delta\mathbf{x} \leq 0 \tag{8}$$

We can rewrite (8) as a quadratic form over a block matrix, as follows:

$$\begin{bmatrix} \delta\mathbf{x}^T & \delta\mathbf{x}^T \end{bmatrix} \begin{bmatrix} (\beta - 2)\mathbf{P} & \mathbf{RDP} \\ \mathbf{PDR} & -2\mathbf{PDP} \end{bmatrix} \begin{bmatrix} \delta\mathbf{x} \\ \delta\mathbf{x} \end{bmatrix} \leq 0 \tag{9}$$

Now the question becomes, when is (9) satisfied? One way to ensure that (9) is satisfied is to ensure that the associated block matrix is always (i.e for all $\mathbf{D}$) negative semi-definite. In that case the inequality will hold over *all* possible vectors, not just $\begin{bmatrix} \delta\mathbf{x} & \delta\mathbf{x} \end{bmatrix}^T$. In other words, the question is now what constraints on the sub-matrices $\mathbf{P}, \mathbf{D}$ and $\mathbf{R}$ ensure that:

$$\forall \mathbf{y} \in \mathbb{R}^{2n}, \quad \mathbf{y}^T \begin{bmatrix} (2 - \beta)\mathbf{P} & -\mathbf{RDP} \\ -\mathbf{PDR} & 2\mathbf{PDP} \end{bmatrix} \mathbf{y} \geq 0 \tag{10}$$

Note that we have multiplied both sides of the inequality by a minus sign. But this is nothing but the definition of a positive semi-definite matrix. Using the Schur complement [Gallier et al., 2020](Proposition 2.1), we know that the block matrix is positive semi-definite iff $\mathbf{PDP} \succ 0$ and:

$$(2 - \beta)\mathbf{P} - \mathbf{RDP}(2\mathbf{PDP})^{-1}\mathbf{PDR} =$$
$$(2 - \beta)\mathbf{P} - \frac{1}{2}(\mathbf{RDR}) \succeq (2 - \beta)\mathbf{P} - \frac{g}{2}(\mathbf{RR}) \succeq 0$$

We continue by setting $\mathbf{P} = \gamma^2 \mathbf{RR}$ with $\gamma^2 = \frac{g}{2(2-\beta)}$, so that the above inequality is satisfied. At this point, we have shown that if $\mathbf{W}$ can be written as:

$$\mathbf{W} = \mathbf{R} - \gamma^2 \mathbf{RR}$$

then (1) is contracting in metric $\mathbf{M} = \gamma^2 \mathbf{RR}$. What remains to be shown is that if the condition:

$$g\mathbf{W} - \mathbf{I} \prec 0$$

Is satisfied, then this implies the existence of an $\mathbf{R}$ such that the above is true. To show that this is indeed the case, assume that:

$$\frac{1}{4\gamma^2}\mathbf{I} - \mathbf{W} \succeq 0$$

Substituting in the definition of $\gamma$, this is just the statement that:

$$\frac{2(2-\beta)}{4g}\mathbf{I} - \mathbf{W} \succeq 0$$

Setting $\beta = 2\lambda > 0$, this yields:

$$(1-\lambda)\mathbf{I} \succeq g\mathbf{W}$$

Since $\mathbf{W}$ is symmetric by assumption, we have the eigendecomposition:

$$\frac{1}{4\gamma^2}\mathbf{I} - \mathbf{W} = \mathbf{V}(\frac{1}{4\gamma^2}\mathbf{I} - \mathbf{\Lambda})\mathbf{V}^T$$

where $\mathbf{V}^T\mathbf{V} = \mathbf{I}$ and $\mathbf{\Lambda}$ is a diagonal matrix containing the eigenvalues of $\mathbf{W}$. Denote the symmetric square-root of this expression as $\mathbf{S}$:

$$\mathbf{S} = \mathbf{V}\sqrt{(\frac{1}{4\gamma^2}\mathbf{I} - \mathbf{\Lambda})}\mathbf{V}^T = \mathbf{S}^T$$

Which implies that:

$$\frac{1}{4\gamma^2}\mathbf{I} - \mathbf{W} = \mathbf{S}^T\mathbf{S}$$

We now define $\mathbf{R}$ in terms of $\mathbf{S}$ as follows:

$$\mathbf{R} = \frac{1}{\gamma}\mathbf{S} + \frac{1}{2\gamma^2}\mathbf{I}$$

Which means that:

$$\frac{1}{4\gamma^2}\mathbf{I} - \mathbf{W} = (\gamma\mathbf{R} - \frac{1}{2\gamma}\mathbf{I})(\gamma\mathbf{R} - \frac{1}{2\gamma}\mathbf{I})$$

Expanding out the right side, we get:

$$\frac{1}{4\gamma^2}\mathbf{I} - \mathbf{W} = \gamma^2\mathbf{R}\mathbf{R} - \mathbf{R} + \frac{1}{4\gamma^2}\mathbf{I}$$

Subtracting $\frac{1}{4\gamma^2}\mathbf{I}$ from both sides yields:

$$\mathbf{W} = \mathbf{R} - \gamma^2\mathbf{R}\mathbf{R}$$

As desired.

$\square$

## C.4 Proof of Theorem 3

**Theorem 3.** *If there exists positive diagonal matrices $\mathbf{P}_1$ and $\mathbf{P}_2$, as well as $\mathbf{Q} = \mathbf{Q}^T \succ 0$ such that*

$$\mathbf{W} = -\mathbf{P}_1\mathbf{Q}\mathbf{P}_2$$

*then (1) is contracting in metric $\mathbf{M} = (\mathbf{P}_1\mathbf{Q}\mathbf{P}_1)^{-1}$.*

*Proof.* Consider again a differential Lyapunov function:

$$V = \delta\mathbf{x}^T\mathbf{M}\delta\mathbf{x}$$

the time derivative is equal to:

$$\dot{V} = -2V + \delta\mathbf{x}^T\mathbf{M}\mathbf{W}\mathbf{D}\delta\mathbf{x}$$

Substituting in the definitions of $\mathbf{W}$ and $\mathbf{M}$, we get:

$$\dot{V} = -2V - \delta\mathbf{x}^T\mathbf{P}_1^{-1}\mathbf{P}_2\mathbf{D}\delta\mathbf{x} \leq -2V$$

Therefore $V$ converges exponentially to zero.

$\square$

## C.5   Proof of Theorem 4

**Theorem 4.** *If $g\mathbf{W} - \mathbf{I}$ is triangular and Hurwitz, then (1) is contracting in a diagonal metric.*

Note that in the case of a triangular weight matrix, the system (1) may be seen as a feedforward (i.e hierarchical) network. Therefore, this result follows from the combination properties of contracting systems. However, our proof provides a means of explicitly finding a metric for this system.

*Proof.* Without loss of generality, assume that $\mathbf{W}$ is lower triangular. This implies that $W_{ij} = 0$ if $i \leq j$. Now consider the generalized Jacobian:

$$\mathbf{F} = -\mathbf{I} + \mathbf{\Gamma}\mathbf{W}\mathbf{D}\mathbf{\Gamma}^{-1}$$

with $\mathbf{\Gamma}$ diagonal and $\Gamma_i = \epsilon^i$ where $\epsilon > 0$. Because $\mathbf{\Gamma}$ is diagonal, the generalized Jacobian is equal to:

$$\mathbf{F} = -\mathbf{I} + \mathbf{\Gamma}\mathbf{W}\mathbf{\Gamma}^{-1}\mathbf{D}$$

Now note that:

$$(\mathbf{\Gamma}\mathbf{W}\mathbf{\Gamma}^{-1})_{ij} = \epsilon^i W_{ij}\epsilon^{-j} = W_{ij}\epsilon^{i-j}$$

Where $i \leq j$, we have $W_{ij} = 0$ by assumption. Therefore, the only nonzero entries are where $i \geq j$. This means that by making $\epsilon$ arbitrarily small, we can make $\mathbf{\Gamma}\mathbf{W}\mathbf{\Gamma}^{-1}$ approach a diagonal matrix with $W_{ii}$ along the diagonal. Therefore, if:

$$\max_i gW_{ii} - 1 < 0$$

the nonlinear system is contracting. Since $\mathbf{W}$ is triangular, $W_{ii}$ are the eigenvalues of $\mathbf{W}$, meaning that this condition is equivalent to $g\mathbf{W} - \mathbf{I}$ being Hurwitz.

$\square$

## C.6   Proof of Theorem 5

**Theorem 5.** *If there exists a positive diagonal matrix $\mathbf{P}$ such that:*

$$g^2\mathbf{W}^T\mathbf{P}\mathbf{W} - \mathbf{P} \prec 0$$

*then (1) is contracting in metric $\mathbf{P}$.*

Note that this is equivalent to the discrete-time diagonal stability condition developed in [Revay and Manchester, 2020], for a constant metric. Note also that when $\mathbf{M} = \mathbf{I}$, Theorem 5 is identical to checking the maximum singular value of $\mathbf{W}$, a previously established condition for stability of (1). However a much larger set of weight matrices are found via the condition when $\mathbf{M} = \mathbf{P}$ instead.

*Proof.* Consider the generalized Jacobian:

$$\mathbf{F} = \mathbf{P}^{1/2}\mathbf{J}\mathbf{P}^{-1/2} = -\mathbf{I} + \mathbf{P}^{1/2}\mathbf{W}\mathbf{P}^{-1/2}\mathbf{D}$$

where $\mathbf{D}$ is a diagonal matrix with $\mathbf{D}_{ii} = \frac{d\phi_i}{dx_i} \geq 0$. Using the subadditivity of the matrix measure $\mu_2$ of the generalized Jacobian we get:

$$\mu_2(\mathbf{F}) \leq -1 + \mu_2(\mathbf{P}^{1/2}\mathbf{W}\mathbf{P}^{-1/2}\mathbf{D})$$

Now using the fact that $\mu_2(\cdot) \leq ||\cdot||_2$ we have:

$$\mu_2(\mathbf{F}) \leq -1 + ||\mathbf{P}^{1/2}\mathbf{W}\mathbf{P}^{-1/2}\mathbf{D})||_2 \leq -1 + g||\mathbf{P}^{1/2}\mathbf{W}\mathbf{P}^{-1/2}||_2$$

Using the definition of the 2-norm, imposing the condition $\mu_2(\mathbf{F}) \leq 0$ may be written:

$$g^2\mathbf{W}^T\mathbf{P}\mathbf{W} - \mathbf{P} \prec 0$$

which completes the proof.

$\square$

## C.7 Proof of Theorem 6

**Theorem 6.** *Let $\mathbf{D}$ be a positive, diagonal matrix with $D_{ii} = \frac{d\phi_i}{dx_i}$, and let $\mathbf{P}$ be an arbitrary, positive diagonal matrix. If:*

$$(g\mathbf{W} - \mathbf{I})\mathbf{P} + \mathbf{P}(g\mathbf{W}^T - \mathbf{I}) \preceq -c\mathbf{P}$$

*and*

$$\dot{\mathbf{D}} - cg^{-1}\mathbf{D} \preceq -\beta\mathbf{D}$$

*for $c, \beta > 0$, then (1) is contracting in metric $\mathbf{D}$ with rate $\beta$.*

*Proof.* Consider the differential, quadratic Lyapunov function:

$$V = \delta\mathbf{x}^T\mathbf{PD}\delta\mathbf{x}$$

where $\mathbf{D} \succ 0$ is as defined above. The time derivative of $V$ is:

$$\dot{V} = \delta\mathbf{x}^T\mathbf{P}\dot{\mathbf{D}}\delta\mathbf{x} + \delta\mathbf{x}^T(-2\mathbf{PD} + \mathbf{PDWD} + \mathbf{DW}^T\mathbf{DP})\delta\mathbf{x}$$

The second term on the right can be factored as:

$$\delta\mathbf{x}^T(-2\mathbf{PD} + \mathbf{PDWD} + \mathbf{DW}^T\mathbf{DP})\delta\mathbf{x} =$$
$$\delta\mathbf{x}^T\mathbf{D}(-2\mathbf{PD}^{-1} + \mathbf{PW} + \mathbf{W}^T\mathbf{P})\mathbf{D}\delta\mathbf{x} \leq$$
$$\delta\mathbf{x}^T\mathbf{D}(-2\mathbf{P}g^{-1} + \mathbf{PW} + \mathbf{W}^T\mathbf{P})\mathbf{D}\delta\mathbf{x} =$$
$$\delta\mathbf{x}^T\mathbf{D}[\mathbf{P}(\mathbf{W} - g^{-1}\mathbf{I}) + (\mathbf{W}^T - g^{-1}\mathbf{I})\mathbf{P}]\mathbf{D}\delta\mathbf{x} \leq$$
$$-cg^{-1}\delta\mathbf{x}^T\mathbf{PD}^2\delta\mathbf{x}$$

where the last inequality was obtained by substituting in the first assumption above. Combining this with the expression for $\dot{V}$, we have:

$$\dot{V} \leq \delta\mathbf{x}^T\mathbf{P}\dot{\mathbf{D}}\delta\mathbf{x} - cg^{-1}\delta\mathbf{x}^T\mathbf{PD}^2\delta\mathbf{x}$$

Substituting in the second assumption, we have:

$$\dot{V} \leq \delta\mathbf{x}^T\mathbf{P}(\dot{\mathbf{D}} - cg^{-1}\mathbf{D}^2)\delta\mathbf{x} \leq -\beta\delta\mathbf{x}^T\mathbf{PD}\delta\mathbf{x} = -\beta V$$

and thus $V$ converges exponentially to 0 with rate $\beta$. $\qquad\square$

## C.8 Proof of Theorem 7

**Theorem 7.** *Satisfaction of the condition*

$$g\mathbf{W}_{sym} - \mathbf{I} \prec 0$$

*is **NOT** sufficient to show global contraction of the general nonlinear RNN (1) in any constant metric. High levels of antisymmetry in $\mathbf{W}$ can make it impossible to find such a metric, which we demonstrate via a $2 \times 2$ counterexample of the form*

$$\mathbf{W} = \begin{bmatrix} 0 & -c \\ c & 0 \end{bmatrix}$$

*with $c \geq 2$.*

Note that $g\mathbf{W}_{sym} - \mathbf{I} = g\frac{\mathbf{W} + \mathbf{W}^T}{2} - \mathbf{I} \prec 0$ is equivalent to the condition for contraction of the system with *linear* activation in the identity metric.

The main intuition behind this counterexample is that high levels of antisymmetry can prevent a constant metric from being found in the nonlinear system. This is because $\mathbf{D}$ is a diagonal matrix

with values between 0 and 1, so the primary functionality it can have in the symmetric part of the Jacobian is to downweight the outputs of certain neurons selectively. In the extreme case of all 0 or 1 values, we can think of this as selecting a subnetwork of the original network, and taking each of the remaining neurons to be single unit systems receiving input from the subnetwork. For a given static configuration of $\mathbf{D}$ (think linear gains), this is a hierarchical system that will be stable if the subnetwork is stable. But as $\mathbf{D}$ can evolve over time when a nonlinearity is introduced, we would need to find a constant metric that can serve completely distinct hierarchical structures simultaneously - which is not always possible.

Put in terms of matrix algebra, D can zero out columns of $\mathbf{W}$, but not their corresponding rows. So for a given weight pair $w_{ij}, w_{ji}$, which has entry in $\mathbf{W}_{sym} = \frac{w_{ij}+w_{ji}}{2}$, if $D_i = 0$ and $D_j = 1$, the $i, j$ entry in $(\mathbf{WD})_{sym}$ will be guaranteed to have lower magnitude if the signs of $w_{ij}$ and $w_{ji}$ are the same, but guaranteed to have higher magnitude if the signs are different. Thus if the linear system would be stable based on magnitudes alone $\mathbf{D}$ poses no real threat, but if the linear system requires antisymmetry to be stable, $\mathbf{D}$ can make proving contraction quite complicated (if possible at all).

*Proof.* The nonlinear system is globally contracting in a *constant* metric if there exists a symmetric, positive definite $\mathbf{M}$ such that the symmetric part of the Jacobian for the system, $(\mathbf{MWD})_{sym} - \mathbf{M}$ is negative definite uniformly. Therefore $(\mathbf{MWD})_{sym} - \mathbf{M} \prec 0$ must hold for all possible $\mathbf{D}$ if $\mathbf{M}$ is a constant metric the system *globally* contracts in with any allowed activation function, as some combination of settings to obtain a particular $\mathbf{D}$ can always be found.

Thus to prove the main claim, we present here a simple 2-neuron system that is contracting in the identity metric with linear activation function, but can be shown to have no $\mathbf{M}$ that simultaneously satisfies the $(\mathbf{MWD})_{sym} - \mathbf{M} \prec 0$ condition for two different possible $\mathbf{D}$ matrices.

To begin, take

$$\mathbf{W} = \begin{bmatrix} 0 & -2 \\ 2 & 0 \end{bmatrix}$$

Note that any off-diagonal magnitude $\geq 2$ would work, as this is the point at which $\frac{1}{2}$ of one of the weights (found in $\mathbf{W}_{sym}$ when the other is zeroed) will have magnitude too large for $(\mathbf{WD})_{sym} - \mathbf{I}$ to be stable.

Looking at the linear system, we can see it is contracting in the identity because

$$\mathbf{W}_{sym} - \mathbf{I} = \begin{bmatrix} -1 & 0 \\ 0 & -1 \end{bmatrix} \prec 0$$

Now consider $(\mathbf{MWD})_{sym} - \mathbf{M}$ with $\mathbf{D}$ taking two possible values of

$$\mathbf{D}_1 = \begin{bmatrix} 1 & 0 \\ 0 & 0 \end{bmatrix} \quad and \quad \mathbf{D}_2 = \begin{bmatrix} 0 & 0 \\ 0 & 1 \end{bmatrix}$$

We want to find some symmetric, positive definite $\mathbf{M} = \begin{bmatrix} a & m \\ m & b \end{bmatrix}$ such that $(\mathbf{MWD}_1)_{sym} - \mathbf{M}$ and $(\mathbf{MWD}_2)_{sym} - \mathbf{M}$ are both negative definite.

Working out the matrix multiplication, we get

$$(\mathbf{MWD}_1)_{sym} - \mathbf{M} = \begin{bmatrix} 2m - a & b - m \\ b - m & -b \end{bmatrix}$$

and

$$(\mathbf{MWD}_2)_{sym} - \mathbf{M} = \begin{bmatrix} -a & -(a + m) \\ -(a + m) & -2m - b \end{bmatrix}$$

We can now check necessary conditions for negative definiteness on these two matrices, as well as for positive definiteness on $\mathbf{M}$, to try to find an $\mathbf{M}$ that will satisfy all these conditions simultaneously. In this process we will reach a contradiction, showing that no such $\mathbf{M}$ can exist.

A necessary condition for positive definiteness in a real, symmetric $n \times n$ matrix $\mathbf{X}$ is $x_{ii} > 0$, and for negative definiteness $x_{ii} < 0$. Another well known necessary condition for definiteness of a real symmetric matrix is $|x_{ii} + x_{jj}| > |x_{ij} + x_{ji}| = 2|x_{ij}| \;\; \forall i \neq j$. See [Weisstein] for more info on these conditions.

Thus we will require $a$ and $b$ to be positive, and can identify the following conditions as necessary for our 3 matrices to all meet the requisite definiteness conditions:

$$2m < a \tag{11}$$

$$-2m < b \tag{12}$$

$$|2m - (a + b)| > 2|b - m| \tag{13}$$

$$|-2m - (a + b)| > 2|a + m| \tag{14}$$

Note that the necessary condition for $\mathbf{M}$ to be PD, $a + b > 2|m|$, is not listed, as it is automatically satisfied if (11) and (12) are.

It is easy to see that if $m = 0$, conditions (13) and (14) will result in the contradictory conditions $a > b$ and $b > a$ respectively, so we will require a metric with off-diagonal elements. To make the absolute values easier to deal with, we will check $m > 0$ and $m < 0$ cases independently.

First we take $m > 0$. By condition (11) we must have $a > 2m$, so between that and knowing the signs of all unknowns are positive, we can reduce many of the absolute values. Condition (13) becomes $a + b - 2m > |2b - 2m|$, and condition (14) becomes $a + b + 2m > 2a + 2m$, which is equivalent to $b > a$. If $b > a$ we must also have $b > m$, so condition (13) further reduces to $a + b - 2m > 2b - 2m$, which is equivalent to $a > b$. Therefore we have again reached contradictory conditions.

A very similar approach can be applied when $m < 0$. Using condition (12) and the known signs we reduce condition (13) to $2|m| + a + b > 2b + 2|m|$, i.e. $a > b$. Meanwhile condition (14) works out to $a + b - 2|m| > 2a - 2|m|$, i.e. $b > a$.

Therefore it is impossible for a single constant $\mathbf{M}$ to accommodate both $\mathbf{D}_1$ and $\mathbf{D}_2$, so that no constant metric can exist for $\mathbf{W}$ to be contracting in when a nonlinearity is introduced that can possibly have derivative reaching both of these configurations. One real world example of such a nonlinearity is ReLU. Given a sufficiently high negative input to one of the units and a sufficiently high positive input to the other, $\mathbf{D}$ can reach one of these configurations. The targeted inputs could then flip at any time to reach the other configuration.

An additional condition we could impose on the activation function is to require it to be a strictly increasing function, so that the activation function derivative can never actually reach 0. We will now show that a very similar counterexample applies in this case, by taking

$$\mathbf{D}_{1*} = \begin{bmatrix} 1 & 0 \\ 0 & \epsilon \end{bmatrix} \quad and \quad \mathbf{D}_{2*} = \begin{bmatrix} \epsilon & 0 \\ 0 & 1 \end{bmatrix}$$

Note here that the $\mathbf{W}$ used above produced a $(\mathbf{WD})_{sym} - \mathbf{I}$ that just barely avoided being negative definite with the original $\mathbf{D}_1$ and $\mathbf{D}_2$, so we will have to increase the values on the off-diagonals a bit for this next example. In fact anything with magnitude larger than 2 will have some $\epsilon > 0$ that will cause a constant metric to be impossible, but for simplicity we will now take

$$\mathbf{W}_* = \begin{bmatrix} 0 & -4 \\ 4 & 0 \end{bmatrix}$$

Note that with $\mathbf{W}_*$, even just halving one of the off-diagonals while keeping the other intact will produce a $(\mathbf{WD})_{sym} - \mathbf{I}$ that is not negative definite. Anything less than halving however will keep the identity metric valid. Therefore, we expect that taking $\epsilon$ in $\mathbf{D}_{1*}$ and $\mathbf{D}_{2*}$ to be in the range $0.5 \geq \epsilon > 0$ will also cause issues when trying to obtain a constant metric.

We will now actually show via a similar proof to the above that $\mathbf{M}$ is impossible to find for $\mathbf{W}_*$ when $\epsilon \leq 0.5$. This result is compelling because it not only shows that $\epsilon$ does not need to be a particularly small value, but it also drives home the point about antisymmetry - the larger in magnitude the antisymmetric weights are, the larger the $\epsilon$ where we will begin to encounter problems.

Working out the matrix multiplication again, we now get

$$(\mathbf{M}\mathbf{W}_*\mathbf{D}_{1*})_{sym} - \mathbf{M} = \begin{bmatrix} 4m - a & 2b - m - 2a\epsilon \\ b - m - 2a\epsilon & -4m\epsilon - b \end{bmatrix}$$

and

$$(\mathbf{M}\mathbf{W}_*\mathbf{D}_{2*})_{sym} - \mathbf{M} = \begin{bmatrix} 4m\epsilon - a & -(2a + m - 2b\epsilon) \\ -(2a + m - 2b\epsilon) & -4m - b \end{bmatrix}$$

Resulting in two new main necessary conditions:

$$|4m - a - b - 4m\epsilon| > 2|2b - m - 2a\epsilon| \tag{15}$$

$$|4m\epsilon - a - b - 4m| > 2|2a + m - 2b\epsilon| \tag{16}$$

As well as new conditions on the diagonal elements:

$$4m - a < 0 \tag{17}$$

$$-4m - b < 0 \tag{18}$$

We will now proceed with trying to find $a, b, m$ that can simultaneously meet all conditions, setting $\epsilon = 0.5$ for simplicity.

Looking at $m = 0$, we can see again that $\mathbf{M}$ will require off-diagonal elements, as condition (15) is now equivalent to the condition $a + b > |4b - 2a|$ and condition (16) is similarly now equivalent to $a + b > |4a - 2b|$.

Evaluating these conditions in more detail, if we assume $4b > 2a$ and $4a > 2b$, we can remove the absolute value and the conditions work out to the contradicting $3a > 3b$ and $3b > 3a$ respectively. As an aside, if $\epsilon > 0.5$, this would no longer be the case, whereas with $\epsilon < 0.5$, the conditions would be pushed even further in opposite directions.

If we instead assume $2a > 4b$, this means $4a > 2b$, so the latter condition would still lead to $b > a$, contradicting the original assumption of $2a > 4b$. $2b > 4a$ causes a contradiction analogously. Trying $4b = 2a$ will lead to the other condition becoming $b > 2a$, once again a contradiction. Thus a diagonal $\mathbf{M}$ is impossible

So now we again break down the conditions into $m > 0$ and $m < 0$ cases, first looking at $m > 0$. Using condition (17) and knowing all unknowns have positive sign, condition (15) reduces to $a + b - 2m > |4b - 2(a + m)|$ and condition (16) reduces to $a + b + 2m > |4a - 2(b - m)|$. This looks remarkably similar to the $m = 0$ case, except now condition (15) has $-2m$ added to both sides (inside the absolute value), and condition (16) has $2m$ added to both sides in the same manner. If $4b > 2(a + m)$ the $-2m$ term on each side will simply cancel, and similarly if $4a > 2(b - m)$ the $+2m$ terms will cancel, leaving us with the same contradictory conditions as before.

Therefore we check $2(a + m) > 4b$. This rearranges to $2a > 2(2b - m) > 2(b - m)$, so that from condition (16) we get $b > a$. Subbing condition (17) in to $2(a + m) > 4b$ gives $8b < 4a + 4m < 5a$ i.e. $b < \frac{5}{8}a$, a contradiction. The analogous issue arises if trying $2(b - m) > 4a$. Trying $2(a + m) = 4b$ gives $m = 2b - a$, which in condition (16) results in $5b - a > |6a - 6b|$, while in condition (17) leads to $5a > 8b$, so (16) can further reduce to $5b - a > 6a - 6b$ i.e. $11b > 7a$. But

$b > \frac{7}{11}a$ and $b < \frac{5}{8}a$ is a contradiction. Thus there is no way for $m > 0$ to work.

Finally, trying $m < 0$, we now use condition (18) and the signs of the unknowns to reduce condition (15) to $a + b + 2|m| > |4b - 2(a - |m|)|$ and condition (16) to $a + b - 2|m| > |4a - 2(b + |m|)|$. These two conditions are clearly directly analogous to in the $m > 0$ case, where $b$ now acts as $a$ with condition (18) being $b > 4|m|$. Therefore the proof is complete. $\qquad\square$

## D  Sparse Combo Net Details

Here we provide comprehensive information on the methodology and results for Sparse Combo Net, including some supplementary experimental results. See the Appendix table of contents for a guide to this section.

### D.1  Extended Methods

#### D.1.1  Initialization and Training

As described in the main text, the nonlinear RNN weights for Sparse Combo Net were randomly generated based on given sparsity and entry magnitude settings, and then confirmed to meet the Theorem 1 condition (or discarded if not). For a sparsity level of $x$ and a magnitude limit of $y$, each subnetwork $\mathbf{W}$ was generated by drawing uniformly from between $-y$ and $y$ with $x\%$ density using scipy.sparse.random, and then zeroing out the diagonal entries. For various potential $x$ and $y$ settings, we quantified both the likelihood that a generated $\mathbf{W}$ would satisfy Theorem 1, and the resulting network performance. Of course this is also dependent on subnetwork size, as larger subnetworks enable greater sparsity. The information we have obtained so far is documented in Section D.2, in particular D.2.2.

In training the linear connections between the described nonlinear RNN subnetworks, we constrained the matrix $\mathbf{B}$ in (3) to reflect underlying modularity assumptions. In particular, we only train the off-diagonal blocks of $\mathbf{B}$ and mask the diagonal blocks. We do this to maintain the interpretation of $\mathbf{L}$ as the matrix containing the connection weights *between* different modules, as diagonal blocks would correspond to self-connections. Furthermore, we only train the lower-triangular blocks of $\mathbf{B}$ while masking the others, to increase training speed.

To obtain the subnetwork RNN metrics necessary for training these linear connections, scipy.integrate.quad was used with default settings to solve for $\mathbf{M}$ in the equation $-\mathbf{I} = \mathbf{MW} + \mathbf{W}^T\mathbf{M}$, as described in the main text. This was done by integrating $e^{\mathbf{W}^T t}\mathbf{Q}e^{\mathbf{W}t}dt$ from 0 to $\infty$. For efficiency reasons, and due to the guaranteed existence of a diagonal metric in the case of Theorem 1, integration was only performed to solve for the diagonal elements of $\mathbf{M}$. Therefore a check was added prior to training to confirm that the initialized network indeed satisfied Theorem 1 with metric $\mathbf{M}$. However, it was never triggered by our initialization method.

Initial training hyperparameter tuning was done primarily with $10 \times 16$ combination networks on the permuted seqMNIST task, starting with settings based on existing literature on this task, and verifying promising settings using a $15 \times 16$ network (Table S2). Initialization settings were held the same throughout, as was later done for the size comparison trials (described in Section D.2.1).

Once hyperparameters were decided upon, the trials reported on in the main text began. Most of these experiments were also done on permuted seqMNIST, where we characterized performance of networks with different sizes and sparsity levels/entry magnitudes. When we moved to the sequential CIFAR10 task, we began by simply training with the same best settings that were found from these experiments. The results of all attempted trials are reported in Section D.4.

Unless specified otherwise, all networks reported on in the main text were trained for 150 epochs, using an Adam optimizer with initial learning rate 1e-3 and weight decay 1e-5. The learning rate was cut to 1e-4 after 90 epochs and to 1e-5 after 140. After identifying the most promising settings, we ran repetitions trials on the best networks for 200 epochs with learning rate cuts after epochs 140 and 190.

### D.1.2 Code and Datasets

All Sparse Combo Nets described in the main text were trained using a single GPU on Google Colab. Code to replicate all experiments can be found here: https://colab.research.google.com/drive/1JCT5OMgaMVK_Xh8BDFNRrEsyFOOjvg10?usp=sharing

Runtime for the best performing architecture settings on the sequential CIFAR10 task was $\sim 24$ hours. A Colab Pro+ account was used to limit timeouts and prioritize GPU access.

The datasets we used for our tasks were MNIST and CIFAR10, downloaded via PyTorch. MNIST is a handwritten digits classification task, consisting of 60,000 training images and 10,000 testing images (each 28x28 and grayscale). It is made available under the terms of the Creative Commons Attribution-Share Alike 3.0 license. CIFAR10 is a dataset of 32x32 color images, split evenly among the following 10 classes: airplane, automobile, bird, cat, deer, dog, frog, horse, ship, and truck. It also contains 60,000/10,000 training/test images, and is distributed under the MIT License.

As mentioned in the main text, we presented these images to our networks pixel by pixel. In the case of MNIST, we used an additional modification to the dataset by permuting the pixels in a randomly determined (but fixed across the dataset) way. The use of these datasets is included in the above link to our code.

Extended results information begins on the next page.

## D.2 Extended Results

In this subsection, we describe additional results we did not get to in the main text, related to scalability, modularity, sparsity, and repeatability. We also add some further discussion of these results, along with more detailed information on the respective experimental set ups.

### D.2.1 Network Size and Modularity Comparison

For the Sparse Combo Net specifically we had additional experiments on architecture size and unit distribution besides what was depicted in the main text. The results of these supplemental experiments both replicated the observed effect in Figure 4 of network size instead using subnetworks with 16 units each as well as higher density (Figure S1A), and evaluated how task performance varies with modularity of a network fixed to have 352 total units (Figure S1B). In the modularity experiment we observed an inverse U shape, with poor performance of a $1 \times 352$ net and an $88 \times 4$ net, and best performance from a $44 \times 8$ net. Note that this experiment compared similar sparsity levels across the different subnetwork sizes. In practice we can achieve better performance with larger subnetworks by leveraging sparsity in a way not possible in smaller subnetworks. These additional experiments are now described in more detail below.

For the initial round of size comparison trials using subnetworks of 16 units each (Figure S1A), the nonlinear RNN weights were set by drawing uniformly from between $-0.4$ and $0.4$ with $40\%$ density using scipy.sparse.random, and then zeroing out the diagonal entries. These settings were chosen because they resulted in $\sim 1\%$ of 16 by 16 weight matrices meeting the Theorem 1 condition. During initialization only the matrices meeting this condition were kept, finishing when the desired number of component RNNs had been set - producing a block diagonal $\mathbf{W}$ like pictured in Figure 3A. This same initialization process was used throughout our experiments. In later experiments we vary the density and magnitude settings.

For the size experiments, we held static the number of units and initialization settings for each component RNN, and tested the effect of changing the number of components in the combination network. 1, 3, 5, 10, 15, 20, 22, 25, and 30 components were tested in this experiment (Figure S1A). Increasing the number of components initially lead to great improvements in test accuracy, but had diminishing returns - test accuracy consistently hit $\sim 93\%$ with a large enough number of subnetworks, but neither loss nor accuracy showed meaningful improvement past the $22 \times 16$ network. Interestingly, early training loss and accuracy became substantially worse once the number of components increased past a certain point, falling from $70\%$ to $43\%$ epoch 1 test accuracy between the $22 \times 16$ and $30 \times 16$ networks. The complete set of results can be found in Table S3.

Note that the size experiment described in the main text (Figure 4A) was a repetition of this original experiment, but now using 32 unit subnetworks with the best performing sparsity settings. The results for the repetition can be found in Table S5.

To better understand how the modularity of the combination networks affects performance, the next experiment held the number of total units constant at 352, selected due to the prior success of the $22 \times 16$ network, and tested different allocations of these units amongst component RNNs. Thus $1 \times 352$, $11 \times 32$, $44 \times 8$, and $88 \times 4$ networks were trained to compare against the $22 \times 16$ (Figure S1B). Increasing the modularity improved performance to a point, with the $44 \times 8$ network resulting in final test accuracy of $94.44\%$, while conversely the $11 \times 32$ resulted in decreased test accuracy. However, the $88 \times 4$ network was unable to learn, and a $352 \times 1$ network would theoretically just be a scaled linear anti-symmetric network.

Because larger networks require different sparsity settings to meet the Theorem 1 condition, these were not held constant between trials in the modularity comparison experiment (Figure S1B), but rather selected in the same way between trials - looking for settings that keep density and scalar balanced and result in $\sim 1\%$ of the matrices meeting the condition. The scalar was applied after sampling non-zero entries from a uniform distribution between -1 and 1. The resulting settings were $7.5\%$ density and 0.077 scalar for 352 unit component RNN, $26.5\%$ density and 0.27 scalar for 32 unit component RNN, $60\%$ density and 0.7 scalar for 8 unit component RNN, and $100\%$ density and 1.0 scalar for 4 unit component RNN. The complete set of results for the modularity experiment can be found in Table S4.

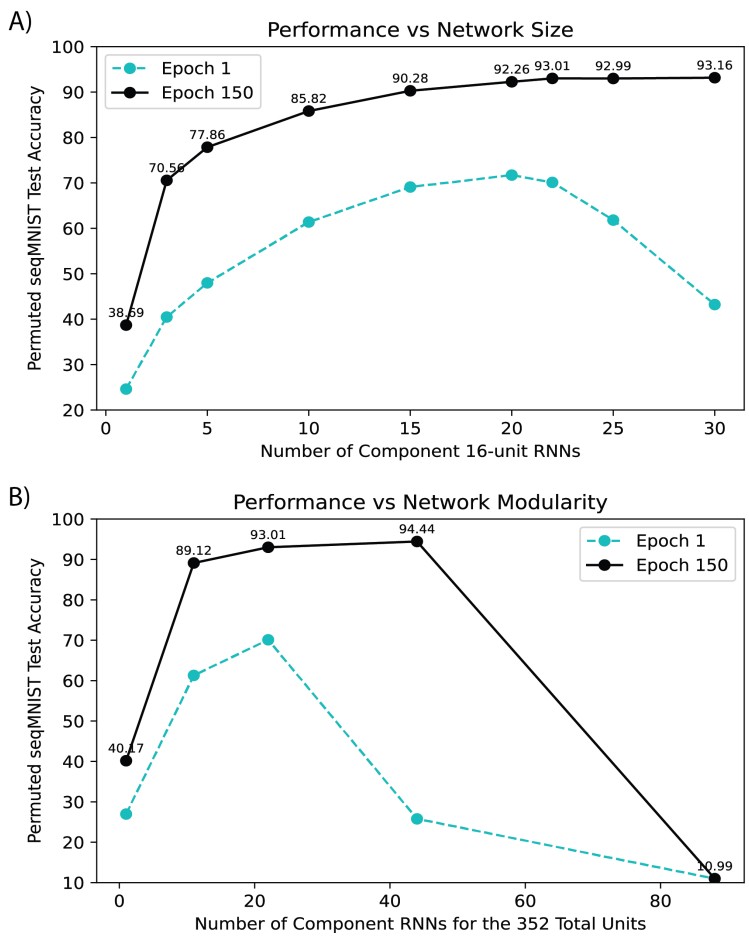

Figure S1: Performance of Sparse Combo Nets on the Permuted seqMNIST task by combination network size. We test the effects on final and first epoch test accuracy of both total network size and network modularity. The former is assessed by varying the number of subnetworks while each subnetwork is fixed at 16 units (A), and the latter by varying the distribution of units across different numbers of subnetworks with the total sum of units in the network fixed at 352 (B). Note that these experiments were run prior to optimizing the sparsity initialization settings. Experiments on total network size were later repeated with the final sparsity settings (Figure 4A). The results of both the size experiments are consistent.

### D.2.2 Sparsity Settings Comparison

Density and scalar settings for the component nonlinear RNNs were initially chosen for each network size using the percentage of random networks that met the Theorem 1 condition. For scalar $s$, a component network would have non-zero entries sampled uniformly between $-s$ and $s$.

When we began experimenting with sparsity in the initialization (after seeing the large performance difference in Figure 5), we split the previously described scalar setting into two different scalars - one applied before a random matrix was checked against the Theorem 1 condition, and one applied after a matrix was selected. Of course the latter must be $\leq 1$ to guarantee stability is preserved. The scalar was separated out after we noticed that at $5\%$ density, random 32 by 32 weight matrices met the condition roughly $1\%$ of the time whether the scalar was 10 or 100000 - $\sim 85\%$ of sampled matrices using scalar 10 would continue to meet the condition even if multiplied by a factor of 10000. Therefore we wanted a mechanism that could bias selection towards matrices that are stable due to their sparsity and not due to magnitude constraints, while still keeping the elements to a reasonable size for training purposes.

Ultimately, both sparsity and magnitude had a clear effect on performance (Figure S2). Increases in both had a positive correlation with accuracy and loss through most parameters tested. Best test accuracy overall was 96.79%, which was obtained by both a $16 \times 32$ network with 5% density and entries between -5 and 5, and a $16 \times 32$ network with 3.3% density and entries between -6 and 6. The latter also achieved the best epoch 1 test accuracy observed of 86.79%. Thus we chose to go with these settings for our extended training repetitions on permuted seqMNIST.

It is also worth noting that upon investigation of the subnetwork weight matrices across these trials, the sparser networks had substantially lower maximum eigenvalue of $|\mathbf{W}|$, suggesting that stronger stability can actually correlate with *improved* performance on sequential tasks. This could be due to a mechanism such as that described in [Radhakrishnana et al., 2020].

Results from the initial trials of different sparsity levels across different network sizes can be found in Table S6. Results from the more thorough testing of different sparsity levels and magnitudes in a $16 \times 32$ network can be found in Table S7.

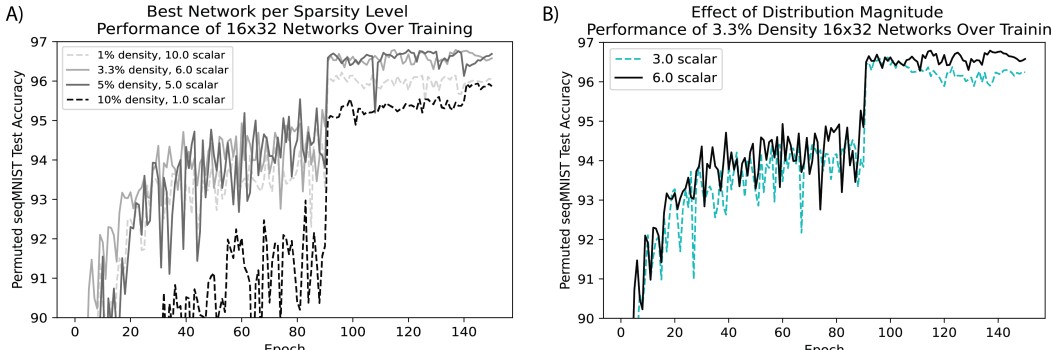

Figure S2: Permuted seqMNIST performance by component RNN initialization settings. Test accuracy is plotted over the course of training for four $16 \times 32$ networks with different density levels and entry magnitudes (A), highlighting the role of sparsity in network performance. Test accuracy is then plotted over the course of training for two 3.3% density $16 \times 32$ networks with different entry magnitudes (B), to demonstrate the role of the scalar. When the magnitude becomes too high however, performance is out of view of the current axis limits.

### D.2.3 Repeatability Tests

Here we give additional details on the repeatability tests described in the main text, as well as introducing some additional results using other hyperparameters (done after the initial SOTA-setting experiments) on sequential CIFAR10.

To further improve performance once network settings were explored on permuted seqMNIST, an extended training run was tested on the best performing option. Settings were kept the same as above using a 3.3% density $16 \times 32$ network, except training now ran for over 200 epochs, with just a single learning rate cut occurring after epoch 200 (exact number of epochs varied based on runtime limit). This experiment was repeated four times and resulted in 96.94% best test accuracy, as described in the main text (Figure S3). Table S8 reports additional details on these trials.

As mentioned, we also characterized Sparse Combo Net performance on the more challenging CIFAR10 sequential image classification task, across a greater number of trials. Ten trials were run for 200 epochs with learning rate scaled after epochs 140 and 190. All other settings were the same as for the permuted seqMNIST repeatability trials. As reported in the main text, the mean test accuracy observed was 64.72%, with variance 0.406, and range 63.73%-65.72%. Training loss and test accuracy over the course of learning are plotted for all ten trials in Figure S4, and exact numbers for each trial are provided in Table S10.

As we encountered difficulties with Colab GPU assignment when we started working on these repetitions, we also trained nine networks over a smaller number of epochs (Figure S5). These networks were trained using the 150 epoch paradigm previously described, although only four of the nine completed training within the 24 hour runtime limit. Complete results for these trials can be

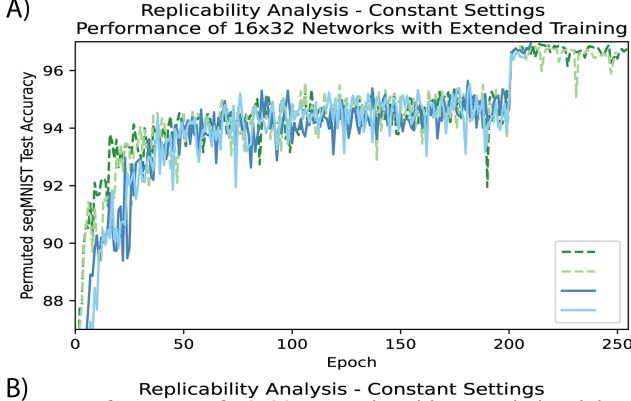

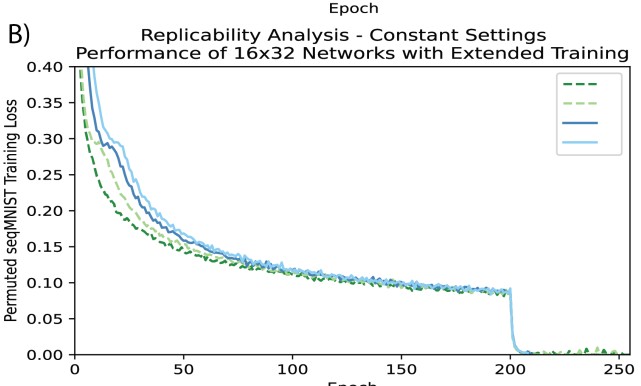

Figure S3: Permuted seqM-NIST performance on repeated trials. Four different $16 \times 32$ networks with 3.3% density and entries between -6 and 6 were trained for 24 hours, with a single learning rate cut after epoch 200. (A) depicts test accuracy for each of the networks over the course of training. (B) depicts the training loss for the same networks. Exact numbers are reported in Table S8.

found in Table S11. Mean performance among the shorter training trials was 62.82% test accuracy with variance 0.95.

Prior to beginning the repeatability experiments on seqCIFAR10, we explored alternative hyperparameters on this task (Table S9). While we ultimately ended up using the same hyperparameters as for permuted seqMNIST, these results further support the robustness of our architecture.

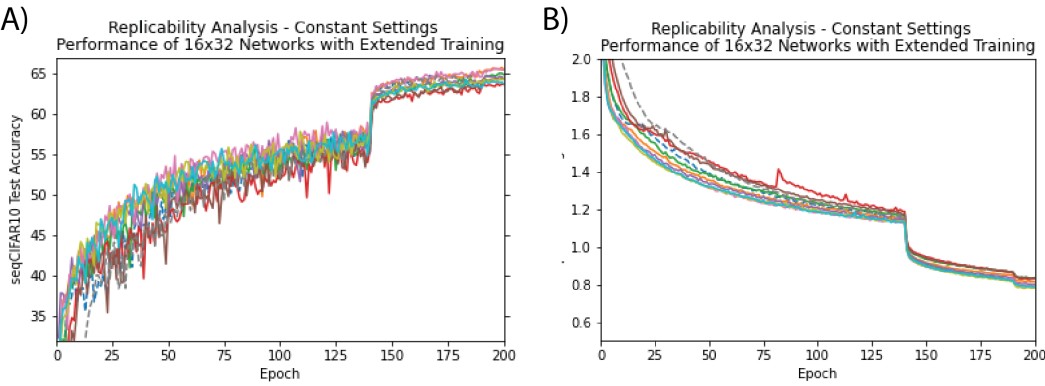

Figure S4: seqCIFAR10 performance on repeated trials. Ten different $16 \times 32$ networks with 3.3% density and entries between -6 and 6 were trained for 200 epochs, with learning rate divided by 10 after epochs 140 and 190. (A) depicts test accuracy for each of the networks over the course of training. (B) depicts the training loss for the same networks. Exact numbers are reported in Table S10.

To complete our benchmarking table, we also ran a single 150 epoch trial of our best network settings on the sequential MNIST task. Test accuracy over the course of training for this trial is depicted in Figure S6.

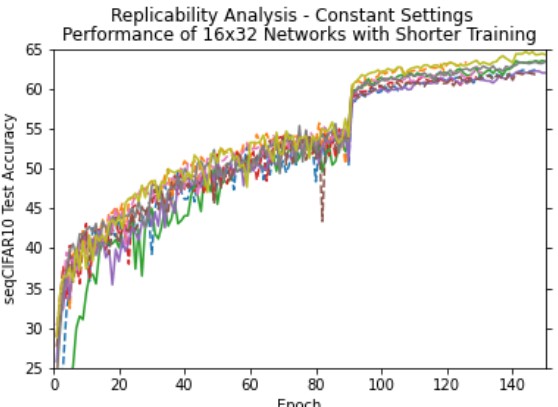

Figure S5: seqCIFAR10 performance on repeated trials with shorter training (done to complete more trials). Nine different $16 \times 32$ networks with 3.3% density and entries between -6 and 6 were set up to train for 150 epochs, with learning rate divided by 10 after epochs 90 and 140. Most of these networks hit runtime limit before completing, however they all got through at least 100 epochs and all had test accuracy exceed 61%. This figure depicts test accuracy for each of the networks over the course of training. Networks that completed training are plotted as solid lines, while those that were cut short are dashed.

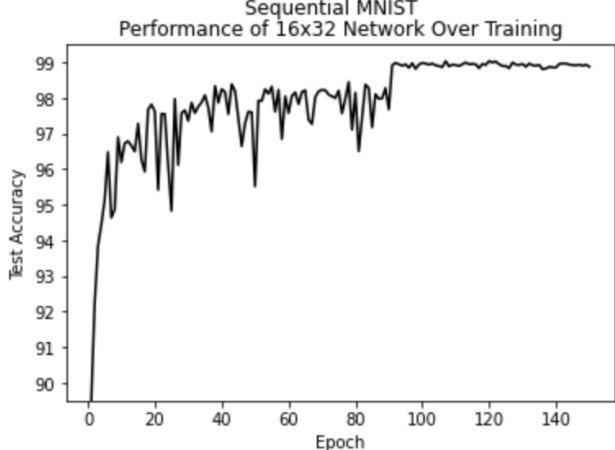

Figure S6: Performance over training on the seqMNIST task for a $16 \times 32$ network with best settings (using 150 epoch training protocol). Final test accuracy exceeded 99%.

### D.2.4 Scalability Pilot: Feedback Sparsity Tests

Not only did the $16 \times 32$ Sparse Combo Net with 3.3% density achieve test accuracy sequential CIFAR10 that is the highest to date for a provably stable RNN, it was higher than the 1 million parameter CKConv network, which set a recent SOTA for permuted seqMNIST accuracy (Table 1). Our network has $130,000$ trainable parameters by comparison. Thus we achieve very impressive results given the characteristics of our architecture.

However, the results of the network size experiments do raise some concern about the scalability of the approach. Here we provide pilot results suggesting that this issue can likely be addressed via the introduction of sparsity in the linear inter-subnetwork connectivity matrix $\mathbf{L}$. While all results in the main text use all-to-all negative feedback, we have early results suggesting that in larger networks, fewer negative feedback connections may perform better, thus preventing performance saturation with scaling.

Below are the results of testing of this idea in a $24 \times 32$ Sparse Combo Net on the sequential CIFAR10 task. We varied the number of feedback connections that were fixed at 0 while all other settings remained static (Table S1). The resulting performance took an inverse U shape, where the network with only 50% of possible feedback connections non-zero had the best test accuracy of 65.14%, achieved in just 124 epochs of training.

| Size | Feedback Density | Epochs | Best Overall Test Acc. | Best Test Acc. Through 85 Epochs |
|---|---|---|---|---|
| 24×32 | 100% | 86 | 52.7% | 52.7% |
| 24×32 | 75% | 88 | 56.49% | 56.48% |
| 24×32 | 66.6% | 89 | 58.84% | 58.84% |
| 24×32 | 50% | 124 | 65.14% | 58.01% |
| 24×32 | 33.3% | 129 | 61.86% | 56.05% |
| 24×32 | 25% | 92 | 54.26% | 50.54% |
| 24×32 | 0% | 130 | 39.8% | 38.38% |
| 16×32 | 100% | 150 | 64.63% | 55.82% |
| 16×32 | 75% | 150 | 64.12% | 57.23% |
| 16×32 | 50% | 127 | 59.87% | 54.26% |

Table S1: Results from pilot testing on the sparsity of negative feedback connections in a $24 \times 32$ Sparse Combo Net and a $16 \times 32$ Sparse Combo Net. Feedback Density refers to the percentage of possible subnetwork pairings that were trained in negative feedback, while the remaining inter-network connections were held at 0. All networks were trained with the same 150 epoch training paradigm as mentioned in the main text, but were stopped after hitting a 24 hour runtime limit. Decreasing Feedback Density is a promising path towards further improving performance as the size of Sparse Combo Nets is scaled. The ideal amount of feedback density will likely vary with the size of the combination network.

## D.3    Architecture Interpretability

In this subsection, we give additional information on trainable parameters and properties of the network weights, to assist in interpreting the Sparse Combo Net architecture and its training process.

### D.3.1    Number of Parameters

To report on the number of trainable parameters, we used the following formula:

$\frac{n^2 - M*C^2}{2} + i*n + n*o + n + o$

Where $n$ is the total number of units in the $M \times C$ combination network, $o$ is the total number of output nodes for the task, and $i$ is the total number of input nodes for the task. Thus for the $16 \times 32$ networks highlighted here, we have 129034 trainable parameters for the MNIST tasks, and 130058 trainable parameters for sequential CIFAR10.

Note that the naive estimate for the number of trainable parameters would be $n^2 + i*n + n*o + n + o$, corresponding to the number of weights in $\mathbf{L}$, the number of weights in the feedforward linear input layer, the number of weights in the feedforward linear output layer, and the bias terms for the input and output layers, respectively. However, because of the combination property constraints on $\mathbf{L}$, only the lower triangular portion of a block off-diagonal matrix is actually trained, and $\mathbf{L}$ is then defined in terms of this matrix and the metric $\mathbf{M}$. Thus we subtract $M * C^2$ to remove the block diagonal portions corresponding to nonlinear RNN components, and then divide by 2 to obtain only the lower half.

### D.3.2    Inspecting Trained Network Weights

After training was completed, we inspected the state of all networks described in the main text, pulling both the nonlinear ($\mathbf{W}$) and linear ($\mathbf{L}$) weight matrices from both initialization time and the final model. For $\mathbf{W}$, we confirmed it did not change over training, and inspected the max real part of the eigenvalues of $|\mathbf{W}|$ in accordance with Theorem 1. The densest tested matrices tended to have $\lambda_{max}(|\mathbf{W}|) > 0.9$, while the sparsest ones tended to have $\lambda_{max}(|\mathbf{W}|) < 0.1$. For $\mathbf{L}$, we checked the maximum element and the maximum singular value before and after training. In general, both went up over the course of training, but by a modest amount.

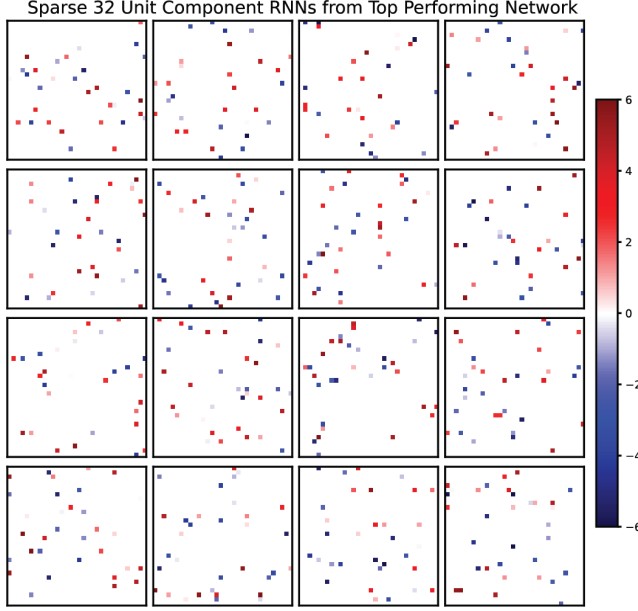

Figure S7: Weight matrices for each of the 32 unit non-linear component RNNs that were used in the best performing $16 \times 32$ network on permuted sequential MNIST.

## D.4 Tables of Results by Trial (Supplementary Documentation)

Table S2 shows all trials run on permuted sequential MNIST before beginning the more systematic experiments reported on in the main text. Notably, our networks did not require an extensive hyperparameter tuning process.

Tables S3 and S4 report additional details on the initial size and modularity experiments (Figure S1). Table S5 reports the results of the repeated experiment on Sparse Combo Net size, this time using 32 unit component subnetworks with best sparsity settings (Figure 4A).

Tables S6 and S7 report results from all trials related to our sparsity experiments (Figures 5 and S2).

Table S8 provides further information on the four trials in the permuted seqMNIST repeatability experiments (Figure S3).

Table S9 reports the results of all trials of different hyperparameters on the sequential CIFAR10 task, in chronological order. Ultimately the same settings as those used for permuted seqMNIST were chosen.

Table S10 shows the results of all ten trials in the seqCIFAR10 repeatability experiments (Figure S4). Table S11 shows results from nine additional seqCIFAR10 trials of shorter training duration (Figure S5), run to increase sample size while unable to access the higher quality Colab GPUs.

Finally, Table S1 reports the results of our pilot trial on introducing sparsity into the linear feedback connection matrix - as this table was presented in Section D.2.4 however, we do not reproduce it here.

### D.4.1 Permuted seqMNIST Trials

| Size | Epochs | Adam WD | Initial LR | LR Schedule | Final Test Acc. |
|---|---|---|---|---|---|
| 10×16 | 150 | 5e-5 | 5e-3 | 0.1 after 91 | 84% |
| 10×16 | 150 | 1e-5 | 1e-2 | 0.1 after 50,100 | 85% |
| 15×16 | 150 | 2e-4 | 5e-3 | 0.1 after 50,100 | 84% |
| 10×16 | 150 | 2e-4 | 1e-2 | 0.5 every 10 | 81% |
| 10×16 | 200 | 2e-4 | 1e-2 | 0.5 after 10 then every 30 | 81% |
| 10×16 | 171* | 5e-5 | 1e-2 | 0.75 after 10,20,60,100 then every 15 | 84% |
| 15×16 | 179* | 1e-5 | 1e-3 | 0.1 after 100,150 | 90% |

Table S2: Training hyperparameter tuning trials, presented in chronological order. * indicates that training was cut short by the 24 hour Colab runtime limit. LR Schedule describes the scalar the learning rate was multiplied by, and at what epochs. The best performing network is highlighted, and represents the training settings we used throughout most of the main text.

| Size | Final Test Acc. | Epoch 1 Test Acc. | Final Train Loss |
|---|---|---|---|
| 1 × 16 | 38.69% | 24.61% | 1.7005 |
| 3 × 16 | 70.56% | 40.47% | 0.9033 |
| 5 × 16 | 77.86% | 47.99% | 0.7104 |
| 10×16 | 85.82% | 61.38% | 0.4736 |
| 15×16 | 90.28% | 69.09% | 0.3156 |
| 20×16 | 92.26% | 71.72% | 0.2392 |
| 22×16 | 93.01% | 70.11% | 0.2073 |
| 25×16 | 92.99% | 61.81% | 0.2017 |
| 30×16 | 93.16% | 43.21% | 0.1991 |

Table S3: Results for combination networks containing different numbers of component 16-unit RNNs. Training hyperparameters and network initialization settings were kept the same across all trials, and all trials completed the full 150 epochs.

| Size | Final Test Acc. | Epoch 1 Test Acc. | Final Train Loss |
|---|---|---|---|
| 1×352 | 40.17% | 26.97% | 1.662 |
| 11×32 | 89.12% | 61.29% | 0.3781 |
| 22×16 | 93.01% | 70.11% | 0.2073 |
| 44 × 8 | 94.44% | 25.78% | 0.1500 |
| 88 × 4 | 10.99% | 10.99% | 2E+35 |

Table S4: Results for different distributions of 352 total units across a combination network. This number was chosen based on prior 22 × 16 network performance. For each component RNN size tested, the same procedure was used to select appropriate density and scalar settings. All networks otherwise used the same settings, as in the size experiments.

| Name | Size | Final Test Acc. |
|------|------|-----------------|
| Sparse Combo Net | $1 \times 32$ | 37.1% |
| Sparse Combo Net | $4 \times 32$ | 89.1% |
| Sparse Combo Net | $8 \times 32$ | 93.6% |
| Sparse Combo Net | $12 \times 32$ | 94.4% |
| Sparse Combo Net | $16 \times 32$ | 96% |
| Sparse Combo Net | $22 \times 32$ | 96.8% |
| Sparse Combo Net | $24 \times 32$ | 96.7% |
| Control | $24 \times 32$ | 47% |

Table S5: Results for Sparse Combo Nets containing different numbers of component 32-unit RNNs with best found initialization settings, using the standard 150 epoch training paradigm. This experiment was run to demonstrate repeatability of the size results seen in Table S3. All trials were run to completion. A control trial was also run with the largest tested network size - the connections between subnetworks were no longer constrained, and thus this control combination network is not certifiably stable.

| Size | Density | Scalar | Final Test Acc. | Epoch 1 Test Acc. | Final Train Loss |
|------|---------|--------|-----------------|-------------------|------------------|
| $11 \times 32$ | 26.5% | 0.27 | 89.12% | 61.29% | 0.3781 |
| $11 \times 32$ | 10% | 1.0 | 94.86% | 70.67% | 0.1278 |
| $22 \times 16$ | 40% | 0.4 | 93.01% | 70.11% | 0.2073 |
| $22 \times 16$ | 20% | 1.0 | 95.27% | 76.58% | 0.0924 |
| $22 \times 16$ | 10% | 1.0 | 94.26% | 71.53% | 0.1425 |
| $44 \times 8$ | 60% | 0.7 | 94.44% | 25.78% | 0.1500 |
| $44 \times 8$ | 50% | 1.0 | 95.05% | 30.52% | 0.1180 |

Table S6: Results for different initialization settings - varying sparsity and magnitude of the component RNNs for different network sizes. All other settings remained constant across trials, using our selected 150 epoch training paradigm.

| Density | Pre-select Scalar | Post-select Scalar | Final Test Acc. | Epoch 1 Test Acc. | Final Train Loss |
|---------|-------------------|--------------------|-----------------|-------------------|------------------|
| 10% | 1.0 | 1.0 | 95.87% | 73.67% | 0.074 |
| 5% | 10.0 | 0.1 | 95.11% | 73.10% | 0.1311 |
| 5% | 10.0 | 0.2 | 96.15% | 82.50% | 0.0051 |
| 5% | 10.0 | 0.5 | 96.69% | 75.76% | 0.0001 |
| 5% | 6.0 | 1.0 | 96.41% | 21.55% | 3.3E-5 |
| 5% | 7.5 | 1.0 | 16.75% | 11.39% | 3068967 |
| 3.3% | 30.0 | 0.1 | 96.24% | 83.89% | 0.0005 |
| 3.3% | 30.0 | 0.2 | 96.54% | 86.79% | 4E-5 |
| 1% | 10.0 | 1.0 | 96.04% | 81.2% | 0.0001 |

Table S7: Further optimizing the sparsity settings for high performance using a $16 \times 32$ network. The final scalar is the product of the pre-selection and post-selection scalars. Note that the 5% density and 7.5 scalar network was killed after 18 epochs due to exploding gradient. All other trials ran for a full 150 epochs.

| Epochs | Best Test Acc. | Epoch 1 Test Acc. | Best Train Loss |
|---|---|---|---|
| 208 | 96.65% | 61.15% | 0.00224 |
| 255 | 96.94% | 84.19% | 5e-5 |
| 250 | 96.88% | 81.08% | 5e-5 |
| 210 | 96.93% | 67.19% | 0.00069 |

Table S8: Repeatability of the best network settings on permuted seqMNIST. Four trials of $16 \times 32$ networks with 3.3% density and entries between -6 and 6, trained for a 24 hour period with a single learning rate cut (0.1 scalar) after epoch 200. All other training settings remained the same as our selected hyperparameters. Trials are presented in chronological order. The mean test accuracy achieved was 96.85% with Variance 0.019.

### D.4.2 seqCIFAR10 Trials

| Density | Pre-select Scalar | Post-select Scalar | Epochs | Adam WD | Initial LR | LR Schedule | Best Test Acc. |
|---|---|---|---|---|---|---|---|
| 3.3% | 30 | 0.2 | 150 | 1e-5 | 1e-3 | 0.1 after 90,140 | 64.63% |
| 3.3% | 30 | 0.2 | 34* | 1e-5 | 5e-3 | 0.1 after 90,140 | 35.42% |
| 5% | 6 | 1 | 150 | 1e-5 | 1e-3 | 0.1 after 90,140 | 60.9% |
| 5% | 10 | 0.5 | 150 | 1e-5 | 1e-4 | 0.1 after 90,140 | 54.86% |
| 3.3% | 30 | 0.2 | 150 | 1e-5 | 5e-4 | 0.1 after 90,140 | 61.83% |
| 3.3% | 30 | 0.2 | 200 | 1e-6 | 2e-3 | 0.1 after 140,190 | 62.31% |
| 3.3% | 30 | 0.2 | 186* | 1e-5 | 1e-3 | 0.1 after 140,190 | 64.75% |
| 3.3% | 30 | 0.2 | 132* | 1e-6 | 1e-3 | 0.1 after 140,190 | 62.31% |
| 5% | 10 | 0.5 | 195* | 1e-5 | 1e-3 | 0.1 after 140,190 | 64.68% |

Table S9: Additional hyperparameter tuning for the seqCIFAR10 task, presented in chronological order. * indicates that training was cut short by the 24 hour Colab runtime limit, or in the case of high learning rate was killed intentionally due to exploding gradient. LR Schedule describes the scalar the learning rate was multiplied by, and at what epochs. The best performing network is highlighted. Ultimately we decided on the same network settings and training hyperparameters for further testing, just extending the training period to 200 epochs with the learning rate cuts occurring after epochs 90 and 140.

| Epochs | Best Test Acc. | Epoch 1 Test Acc. | Best Train Loss |
|---|---|---|---|
| 186 | 64.75% | 10.53% | 0.83 |
| 200 | 64.04% | 26.35% | 0.787 |
| 200 | 64.32% | 26.76% | 0.778 |
| 170 | 64.88% | 20.65% | 0.857 |
| 200 | 65.72% | 27.28% | 0.813 |
| 200 | 63.73% | 10.47% | 0.826 |
| 200 | 65.03% | 18.75% | 0.83 |
| 200 | 64.71% | 32.36% | 0.799 |
| 200 | 64.4% | 10.09% | 0.83 |
| 200 | 65.63% | 30.25% | 0.792 |

| Epochs | Best Test Acc. | Epoch 1 Test Acc. | Best Train Loss |
|---|---|---|---|
| 150 | 64.63% | 28.91% | 0.837 |
| 150 | 63.51% | 9.7% | 0.9 |
| 148 | 62.35% | 17.51% | 0.89 |
| 126 | 63.33% | 23.61% | 0.903 |
| 129 | 61.45% | 28.93% | 0.937 |
| 150 | 62.21% | 25.82% | 0.9 |
| 150 | 63.42% | 23.51% | 0.86 |
| 147 | 62.05% | 27.67% | 0.882 |
| 131 | 62.41% | 30.33% | 0.912 |

Table S10: Repeatability of the best network settings on seqCIFAR10. Ten trials of $16 \times 32$ networks with 3.3% density and entries between -6 and 6, trained for 200 epochs with learning rate scaled by 0.1 after epochs 140 and 190. All other training settings remained the same as before. Trials are presented in chronological order. The mean test accuracy achieved was 64.72% with Variance 0.406. Most trials completed all 200 epochs, but two were cut short due to runtime limits.

Table S11: An additional nine trials investigating the repeatability of our results on the seqCIFAR10 task. For these trials we used the same 150 epoch training paradigm as previously, although only four of the networks were able to fully complete training. These trials were done to expand our sample size while access was limited to only slower GPUs. The mean observed test accuracy among the shorter trials was 62.82%, with variance of 0.95.

# E    SVD Combo Net Details

## E.1    Parameterization Information

For the SVD Combo Net, we ensured contraction by directly parameterizing each of the $\mathbf{W}_i$ ($i = 1, 2 \ldots p$) as:

$$\mathbf{W}_i = \mathbf{\Phi}_i^{-1} \mathbf{U}_i \mathbf{\Sigma}_i \mathbf{V}_i^T \mathbf{\Phi}_i \tag{19}$$

where $\mathbf{\Phi}_i$ is diagonal and nonsingular, $\mathbf{U}_i$ and $\mathbf{V}_i$ are orthogonal, and $\mathbf{\Sigma}_i$ is diagonal with $\Sigma_{ii} \in [0, g^{-1})$. We ensure orthogonality of $\mathbf{U}_i$ and $\mathbf{V}_i$ during training by exploiting the fact that the matrix exponential of a skew-symmetric matrix is orthogonal, as was done in [Lezcano-Casado and Martınez-Rubio, 2019]. The network constructed from these subnetworks using (2) is contracting in metric $\tilde{\mathbf{M}} = \text{BlockDiag}(\mathbf{\Phi}_1^2, \ldots, \mathbf{\Phi}_p^2)$.

## E.2    Control Experiment

To do the SVD Combo Network stability control experiment, we still constrained the weights of the subnetwork modules using the above parameterization, but we removed any constraint on the connections between modules. Thus each individual module was still guaranteed to be contracting, but the overall system had no such guarantee in this control trial. The experiment was run using a $24 \times 32$ combination network and the permuted sequential MNIST task.

In contrast to the primary control experiment run on Sparse Combo Net, for the SVD Combo Net we saw only a very slight performance decrease when the between-module connections were no longer constrained for the overall stability certificate. Test accuracy in the control run was 94.56%, as compared to the 94.9% observed with the standard SVD Combo Net.

This disparity in performance decrease makes sense when considering that the hidden-to-hidden weights of SVD Combo Network are trainable, while those of the Sparse Combo Network are not.

Whatever instabilities are introduced by the lack of constraint on the inter-subnetwork connections likely cannot be adequately compensated for in the Sparse Combo Network.

## E.3 Model Code

```python
#define network
class rnnAssembly_SV(LightningModule):
    '''
    Pytorch module for training the following system:
        tau*dx/dt = -x + W*phi(x) + L*x+ u(t)
    where tau > 0, phi is a nonlinearity, W is block diagonal, L is some 'contracting' combination matrix and u is some input.

    '''

    def __init__(self, input_size, hidden_sizes, output_size,alpha,A):
        super(rnnAssembly_SV, self).__init__()
        self.input_size = input_size
        self.hidden_sizes = hidden_sizes
        self.hidden_size = int(np.sum(hidden_sizes))
        ns = hidden_sizes
        self.output_size = output_size
        self.alpha = alpha

        #input to hidden weights
        self.weight_ih = nn.Parameter(torch.normal(0,1/np.sqrt(self.hidden_size),(self.hidden_size, self.input_size)))
        #hidden to output weights
        self.weight_ho = nn.Parameter(torch.normal(0,1/np.sqrt(self.hidden_size),(self.output_size, self.hidden_size)))

        #mask to modularize overall network
        self.register_buffer("V_mask", create_mask_given_A(torch.eye(len(ns)),ns).bool())

        #parameterization of hidden-to-hidden networks
        self.U = nn.Parameter(torch.eye(self.hidden_size))
        self.V = nn.Parameter(torch.eye(self.hidden_size))
        self.Sigma = nn.Parameter(torch.ones((self.hidden_size,)))
        self.Phi = nn.Parameter(torch.normal(1,1/np.sqrt(self.hidden_size),(self.hidden_size,)))
        self.sing_val_eps = 1e-3
        self.register_buffer("L_mask", create_mask_given_A(A,ns).bool())
        self.register_buffer("S_offset",(self.sing_val_eps)*torch.ones(self.Sigma.shape[0]))

        #trainable inter-subnetwork weights
        self.L_train = nn.Parameter(self.L_mask*torch.normal(0,1/np.sqrt(np.mean(ns)),(np.sum(ns), np.sum(ns))))

        #biases
        self.bias_oh = nn.Parameter(torch.normal(0,1/np.sqrt(self.hidden_size),(1,self.output_size)))
        self.bias_hh = nn.Parameter(torch.normal(0,1/np.sqrt(self.hidden_size),(1,self.hidden_size)))

    def forward(self, input):

        #mask the left and right orth 'stem' matrices to produce diagonal-block structure
        U_masked = self.U*self.V_mask
        V_masked = self.V*self.V_mask

        #take skew-symmetric part of these matrices
        #exponentiate skew-symmetric part to produce left and right orthogonal matrices
        O1 = torch.matrix_exp(U_masked - U_masked.T)
        O2 = torch.matrix_exp(V_masked - V_masked.T)

        #get diagonal of singular value matrix, with elements between [0,1)
        S = torch.exp(-self.Sigma**2 -self.S_offset)

        #construct the weight matrix to be contracting in metric determined by Phi
        W = torch.diag_embed(self.Phi) @ (O1 @ torch.diag_embed(S) @ O2.T) @ torch.diag_embed(1/self.Phi)

        #get metric and metric inverse
        M = torch.diag_embed(1/(self.Phi**2))
        M_inv = torch.diag_embed(self.Phi**2)

        #get mask for network-to-network coupling
        L_masked = self.L_train*self.L_mask

        #initialize hidden state of the network
        inputs = input.unbind(1)
        state = torch.zeros((input.shape[0],self.hidden_size),device = self.device)
        state = state.type_as(state)

        #propagate input through the dynamics and store outputs
        for i in range(len(inputs)):
            fx = -state + F.relu(state @ W.T + inputs[i] @ (self.weight_ih.T) + self.bias_hh) + state @ (L_masked.T - (M @ L_masked) @ M_inv)
            state = state + self.alpha*fx
        hy = state @ (self.weight_ho.T)
        return hy, state
```

Figure S8: Pytorch Lightning code for SVD Combo Net cell.