# OpenReview forum: "RNNs of RNNs: Recursive Construction of Stable Assemblies of Recurrent Neural Networks"
_NeurIPS.cc/2022/Conference — NeurIPS 2022 Accept_

### Official Review · Reviewer_3ViV · 2022-06-19

**Rating:** 6
**Confidence:** 4
**Soundness:** 3 good
**Presentation:** 3 good
**Contribution:** 3 good

**Summary:**

With the increasing ability of experimental neuroscientists to obtain simultaneous recordings of brain activity from multiple areas of the brain, it is important to develop computational theory for RNNs that can model such situations. In this context, the submission is focused on the question of the stability of networks of multiple RNNs that are connected and in particular the question of contractive stability. The paper presents a network of networks model that involves multiple continuous time rate-based vanilla RNNs whose state updates depend on the states of the other RNNs.  A parameterization of negative feedback between the multiple RNNs is presented which allows for optimization using gradient-based techniques and it is proved that this ensures contractive stability of the combined system (assuming each individual RNN is contracting). In addition, multiple sufficient conditions are presented to ensure contractive stability of the individual RNNs.

Two particular instantiations of the networks of networks model based on two of the stability results are empirically evaluated on the seqMNIST, permuted seqMNIST and seqCIFAR sequence classification tasks to determine if the stability constraints impede expressivity. One of the variants outperforms other recent stable RNNs on both permuted SeqMNIST and seqCIFAR and outperforms an LSTM baseline on all three. In addition, experiments are performed to study how the performance increases as more networks are added and the sparsity is adjusted.



**Questions:**

1. There seems to be a close connection with multidimensional RNNS (Graves 2007) and multidimensional state space models (https://en.wikipedia.org/wiki/Multidimensional_system), where the update equations for each of the multiple states of the system depend on the other states. Perhaps a more detailed discussion of this connection would be useful for the readers and potentially broaden the reach of this work? There could be relevant ideas in the multidimensional systems theory literature as well.

2. Lines 74-75 say that later on extensions to nonlinear couplings will be discussed. This sounded interesting, but unless I am missing something, the only mention of this I saw was the last sentence of line 107, which I am not sure I would consider a discussion.  I would recommend either rewording lines 74-75 where you are not overpromising or expanding the discussion (or referring to its location more explicitly if I am missing it somewhere else).

3. The freezing of the weights of the Sparse Combo Net has a connection to Echo State Networks which might also be included in a related works section.

4. Why are the results for the SVD Combo net not presented in the results Table 1? It seems strange to specifically discuss this network and then not present the results. Some of the results (though I do not think all) appear to be in the text and appendix, but it seems including this in the table would reduce confusion.

5. Why were only the two variants determined by Theorem 1 and 5  (Sparse combo net and SVD combo net) considered instead of other variants based on the other theorems as well? Unless I missed this discussion in the appendix, a more detailed discussion of this choice might be helpful.

6. Section 4.2.1 discusses adjusting the network size. Increasing the number of modules increases the number of parameters, so one natural question would be how does the performance of 2 size 32 modules compare to the performance of 1 size 64 module, and so on...? It is somewhat unclear to me what this would look like for the Sparse combo net since many of the parameters are not trained, but it seems natural for the SVD combo net.

Minor points:

-Line 194-195 says "only trained the connections between subnetworks (Figure 2A)" but I think this should say Figure 2B (the sparse combo net)

-Line 195 says "we trained all parameters of the model (Figure 2B)" but I think this should be Figure 2C (the SVD Combo net)

-There are few places where being a bit more explicit about the shapes of some of the variables might help. For example, in Corollary 1 in the appendix, I think you would benefit from more explicitly stating that L is a matrix made up of matrices (perhaps explicitly providing the general dimensions) where each $L_{ij}$ is a matrix. Similarly, explicitly saying u(t) is made up of $u_i(t)$ could be helpful since in the main text u(t) is only defined for eq. 1 (a single RNN). Alternatively, you could maybe define $\tilde{u}(t)$ analogously to $\tilde{x}$.

**Limitations:**

The authors discuss limitations in an extended discussion in the appendix.

**Strengths And Weaknesses:**

-Originality: While the proposed network of RNNs appears to fall within the class of multidimensional RNNs and multidimensional state space models (though see Question #1 below for clarification on this point) and the idea of using negative feedback to ensure stability of combined systems is well-known in the nonlinear systems literature, this appears to be the first work to study in detail how to parameterize these stability conditions in a way that is straightforward to optimize using deep learning techniques. In addition, the conditions derived to ensure contractive stability of a single RNN appear to be novel. It seems however that the paper would benefit from a more detailed related works section to help the reader have a better survey of where this work fits into the literature.

-Quality: The theoretical results appear to be sound. The empirical results seem to support the basic claim that some of the theoretical stability conditions can be used in a way that does not severely inhibit the expressivity of the network. However, I have some lingering questions related to these results that could be clarified. See Questions section below. In addition, the major motivation for the approach in the paper is for modeling neuroscience data recorded from multiple brain areas, so some kind of modeling experiment related to this (even if synthetic) would strengthen the submission.

-Clarity: The paper is in general well-written and clear. See Questions sections below for a few detailed suggestions for improvement.

-Significance: This paper will be of interest to the computational neuroscience community as well as the general machine learning community interested in the stability of RNNs.  This paper seems to have connections to multidimensional RNNs and a more thorough discussion of this connection could broaden the reach to people interested in modeling multidimensional data.  The theoretical stability results could help guide future development of these methods. In addition, in my experience seqCIFAR is still a challenging task for most methods and good performance on this task tends to translate to good performance on other tasks requiring the modeling of long-range dependencies. So the fact that these combined networks perform better than many of the baseline methods is quite interesting.

---

> ### Author Response · Authors · 2022-07-31
> **Thank you for your positive and careful review of our paper. We address your comments below**
>
> > There seems to be a close connection with multidimensional RNNS...Perhaps a more detailed discussion of this connection would be useful for the readers and potentially broaden the reach of this work?
>
> We thank the reviewer for bringing this literature to our attention, we were not previously aware of it. We agree that these ideas are related–indeed, perhaps complementary. A main difference between our approach and the approach described in Graves, 2007, is that in the latter, an increase in the number of sequence dimensions is accommodated via an increase in the amount of recurrence per neuron. Our RNNs do not have this property. However, we see no reason a priori why our “RNNs of RNNs” cannot also be “RNNs of Multidimensional RNNs”. The contraction constraints on the hidden-to-hidden weight matrices will almost certainly have to be adjusted from our current setting, but this is an interesting direction of research.
> > Lines 74-75 say that later on extensions to nonlinear couplings will be discussed...
>
> We have edited the manuscript to reflect that we only consider nonlinear hierarchical couplings (old manuscript, Lines L101 and L544). It is an open and interesting question on how to include nonlinear negative feedback connections—something we are actively working on.
> We have also edited the manuscript to note that linear couplings are a valid approximation to nonlinear couplings around fixed points, in the same way that linear RNNs are valid approximations to nonlinear RNNs around fixed points.
>
> > Why are the results for the SVD Combo net not presented in the results Table 1?
>
> The SVD Combo Net never reached above 55% test accuracy for CIFAR10 in our early experiments, while the Sparse Combo Net easily surpassed that. On the basis of this finding, we decided to only conduct thorough analyses on the Sparse Combo Net. Additionally, in Table 1 we prioritized networks that established important baselines or were SOTA in a given category. As we are comparing only the best performing network variants from an already carefully selected subset of papers in this table, we felt our architecture that obtained the best accuracy would be the relevant comparison. We have clarified this point in the revised manuscript.
>
> >Why were only the two variants determined by Theorem 1 and 5 (Sparse combo net and SVD combo net) considered instead of other variants based on the other theorems as well? Unless I missed this discussion in the appendix, a more detailed discussion of this choice might be helpful.
>
> We chose these two architectures because they represent two distinct ways of training “RNNs of RNNs”: 1) training only the interareal weights and 2) training all the weights in the network. We chose the sparsity condition because we were motivated by neuroscience: cortical connectivity is known to be extremely sparse. We chose the SVD condition, because it (in a less general form) has been explored in prior work (Jaeger  2001, Miller and Hardt 2018, Revay and Manchester, PMLR, 2020). We plan to explore the performance of all these conditions across more tasks in future work, where we do not have to “compete” for space with our novel theoretical findings.
>
> >...how does the performance of 2 size 32 modules compare to the performance of 1 size 64 module, and so on...?
>
> This is also something we were interested in, which was done as an early experiment but later moved to the supplement (section A4.2.1). For the Sparse Combo Net we ran an experiment on permuted sequential MNIST where we held the total sum of units in the network fixed at 352, but varied the number of modules these units were spread over. With all 352 units in 1 module the test performance was ~40%, which would be attributable to training of the linear feedforward input and output layers, because as you mention there is no weight updating of the RNN in this case. With 4 units each in 88 RNN modules the network was unable to learn at all, suggesting that a pure linear feedback network would be unable to do the task. The other tested modularities (11 RNNs, 22 RNNs, and 44 RNNs) all had test performance around 90% or better - see Figure S1(B) for further results.
>
> > There are few places where being a bit more explicit about the shapes of some of the variables might help...
>
> We have defined $\tilde{u}(t)$, and included the dimensions of the submatrices $\mathbf{L}_{ij}$ in the manuscript.

---

> > ### Comment · Reviewer_3ViV · 2022-08-06
> > **Thank you and follow up question**
> >
> > Thank you for the detailed response. Many of my questions have been resolved. However, I would like to ask for further clarification on one point.
> >
> > In your response,  you say:  *"A main difference between our approach and the approach described in Graves, 2007, is that in the latter, an increase in the number of sequence dimensions is accommodated via an increase in the amount of recurrence per neuron. Our RNNs do not have this property. However, we see no reason a priori why our “RNNs of RNNs” cannot also be “RNNs of Multidimensional RNNs”."*
> >
> > Could you elaborate further on this point? In my mind, eq. 2 of your paper is a multidimensional RNN or multidimensional state-space model (SSM) where the overall system state matrix has the different $\mathbf{W}$ matrices on the diagonal and the different $\mathbf{L}$ matrices on the off-diagonals .
> >
> > I realize now that the reference I provided is somewhat unclear since it does not provide explicit equations.  See https://en.wikipedia.org/wiki/Multidimensional_system for example equations for the linear multidimensional SSM case, which seem to match the equations you wrote in your follow-on response to reviewer HWhw.
> >
> > To be clear I really like the main ideas of your work and do not think this point detracts from your contributions. However, I do think there is a connection to be made with the existing literature on multidimensional systems theory.  I also have to admit that I personally find the "RNN of RNNs" terminology perhaps less clear than you intend and think perhaps other readers may find this multidimensional system view helpful.
> >
> > Alternatively, perhaps I am missing something about your approach and would like to better understand if so.

---

> > > ### Author Response · Authors · 2022-08-08
> > > **Thank you for the follow-up question. We agree with your broader point, but draw a distinction with the Graves (2007) paper below:**
> > >
> > > After digging into the multidimensional state space literature more closely, we now agree with you: our networks can indeed be viewed as a special case of a multi-dimensional state space model. We will update the manuscript to point out this connection.
> > >
> > > With regards to the 2007 Graves paper, our networks have more in common with the Graves work on ["Deep RNNs"](https://arxiv.org/pdf/1303.5778.pdf) from 2013, in which the authors stack RNNs on top of one another. In the language of our paper, this would correspond to a particular hierarchical combination type. As far as we can tell, the distinction between this "Deep RNN" approach and the "Multidimensional RNN" (MDRNN) approach is as follows: in the MDRNN approach each dimension of the input space is given a corresponding hidden state in the network. This enables each dimension of the input sequence to be processed simultaneously. This is in contrast to the Deep RNN approach, where the input has to be manipulated into a one-dimensional sequence in order to be passed into the network.
> > >
> > > As a concrete example, this is the difference between passing each column of an image (or pixel)  into the RNN one at a time, as done in Deep RNNs and our networks, and passing in the whole image at once (MDRNN). Therefore the difference between our networks and the MDRNN approach has more to do with how the inputs are passed into the network than the actual network architecture itself.
> > >
> > > If you agree with this characterization, we will also include a discussion of it into our paper, as we agree that it is a useful connection to pre-existing literature.

---

> > > > ### Comment · Reviewer_3ViV · 2022-08-08
> > > > **Thank you**
> > > >
> > > > Thanks for your reply. I only mentioned the Graves 2007 paper since people often refer to this for the multidimensional RNN setting. I personally think the connection with the more general multi-dimensional state space literature is sufficient and more important. In addition, it is actually not clear to me how relevant the Graves 2013 paper is, so I will leave that up to you if you think it is useful to include.
> > > >
> > > > The broader point is that Eq. 2 appears to represent a multi-dimensional state space model, where each neural population is assigned a state (and can thus be thought of as a different "dimension" of the multidimensional system.) Each neural population (or dimension) both evolves on its own and also interacts with the other neural populations (or dimensions of the multidimensional system). It seems that some of the research into stability in this literature may be relevant to your work, and vice versa.
> > > >
> > > > Thank you for your response!

---

### Official Review · Reviewer_HWhw · 2022-07-08

**Rating:** 5
**Confidence:** 3
**Soundness:** 3 good
**Presentation:** 2 fair
**Contribution:** 2 fair

**Summary:**

This paper tries to address the stability issue of combining multiple stable RNNs. The motivation is that the stability of a single RNN is well studied, but the stability of this kind “RNNs of RNNs” is little studied, particularly, as compared to the recent rising research direction in neuroscience. The “RNNs of RNNs” is defined in eq.(2), which considers the simplest case: linear combination.  The authors provided some theoretical justification to prove that the global stability of “RNNs of RNNs” can be achieved. The authors tested one instance network: stability constrained network on sequential MNIST and Cifar10, showed that good performance can be achieved as compared to baselines.

**Questions:**

1. What is the "stability" of RNN meaning? it should be given and explained in a better way at the beginning of the paper. This causes confusion in the later stage.
2. The term "RNNs of RNNs" is confusing, in eq (2), the connection between RNNs is not recurrent/recursive, but linear connection.
3. Given the linear combination of RNNs, what is the difference to ensemble RNNs?
4. How the empirical results support the main claims? Good accuracy presents stability?

5.There are many methods give better performance on mentioned datasets, just give a few examples:
[1]Chang et al, Dilated Recurrent Neural Networks, NeuriPS 2017.

[2] Tim Cooijmans et al, Recurrent Batch Normalization, ICLR 2017.

[3] Wang et al, State-Regularized Recurrent Neural Networks, ICML 2019.

[4] Trinh et al, Learning longer-term dependencies in rnns with auxiliary losses, ICML 2018.


**Limitations:**

yes

**Strengths And Weaknesses:**

Strengths:
1. The core idea of studying RNN contracting stability is interesting as well as the connection to neuroscience.
2. Some theoretical findings would be helpful in understanding the “network of network” stability.

Weaknesses:
1. There are some unclear points, which makes this paper become less and less understandable in the later stage.
2. The experimental section is weak. Experiments on harder tasks such as language modelling, text classification, would make this section much stronger.

---

> ### Author Response · Authors · 2022-07-31
> **We thank the reviewer for their comments, questions, and positive assessment.  We have addressed several of their points, but would appreciate clarification on a few others**
>
> > What is the "stability" of RNN meaning? it should be given and explained in a better way at the beginning of the paper. This causes confusion in the later stage.
>
> Throughout our paper, stability means “contractive stability”. We provide a brief primer on contraction in L38-47, as well as A1.2. Contraction is a strong form of exponential stability, which implies many other weaker forms of stability (for example input-to-state stability).
> We have incorporated your comments into the manuscript by expanding the introductory section to better explain what contractive stability is.
>
> > The term "RNNs of RNNs" is confusing, in eq (2), the connection between RNNs is not recurrent/recursive, but linear connection.
>
> We would appreciate a clarification by what you mean here. The connection between RNNs in our framework is both recurrent and linear (these terms are not mutually exclusive). For example, a simple linear RNN given by x(t+1) = Ax(t) is both recurrent and linear, except for special A matrices.
> > Given the linear combination of RNNs, what is the difference to ensemble RNNs?
>
> We are unfortunately unfamiliar with “ensemble RNNs”, and a Google search did not appear to reveal a single, well-defined concept. We would greatly appreciate a link or description of what you mean by “ensemble RNN”, so that we can better answer your question. We would be interested to learn more about this architecture–perhaps our stability results could be useful there as well.
>
> > How the empirical results support the main claims? Good accuracy presents stability?
>
> Our empirical results support our claims in two ways. The first way is that we claim our stability constrained feedback combinations are optimizable using deep learning. We empirically show that this is true, by training these stability-constrained RNNs on challenging tasks. The second way is that our empirical results support our claim that stability-constrained RNNs of RNNs are potentially useful for neuroscience, by showing our stability constraints are not too restrictive, and the stability-constrained RNNs can perform challenging tasks.
>
> >There are many methods give better performance on mentioned datasets, just give a few examples…
>
> As explained in the main text (e.g old manuscript L54-55), our claims of SOTA only apply to the model class of provably stable recurrent architectures. We also cite some of the papers you provide in our original manuscript (e.g Trinh 2018), in addition to other papers which outperform our models of the benchmarks we considered. Importantly, however, none of these models are guaranteed to be stable. Stability guarantees are necessary in the machine learning context for reasons of safety and robustness (see e.g Revay & Manchester, 2020, Proceedings of Machine Learning Research).

---

> > ### Comment · Reviewer_HWhw · 2022-08-03
> > **Thanks for the response and further questions**
> >
> > 1. It is true that x(t+1) = Ax(t) is both recurrent and linear, but the term "RNNs of RNNs" gives me an impression that x(t) is a RNN, is this true from the paper? If the x(t) is the output from another RNN, what is difference to stacked RNNs, or other ways of combining RNNs.
> >
> > 2. “ensemble RNNs” , means ensembles of RNNs. https://en.wikipedia.org/wiki/Ensemble_learning
> >
> > 3. "however, none of these models are guaranteed to be stable", how do we know other methods are not contractive stable?

---

> > > ### Author Response · Authors · 2022-08-04
> > > **Thank you for clarifications and additional questions. We respond below:**
> > >
> > > > It is true that x(t+1) = Ax(t) is both recurrent and linear, but the term "RNNs of RNNs" gives me an impression that x(t) is a RNN, is this true from the paper? If the x(t) is the output from another RNN, what is difference to stacked RNNs, or other ways of combining RNNs.
> > >
> > > In our networks, the output from another RNN is passed through another set of weights before entering a downstream RNN. This is slightly different from the equation you have written above. For two linear RNNs, our network equations would read:
> > >
> > > $\dot{\mathbf{x}} = \mathbf{A}\mathbf{x}(t) + \mathbf{C}\mathbf{y}(t)$
> > > and
> > > $\dot{\mathbf{y}} = \mathbf{B}\mathbf{y}(t) - \mathbf{C}^T\mathbf{x}(t)$
> > >
> > > Our approach differs from other ways of combining RNNs by carefully constraining the connection matrices between RNNs (matrix $C$ in the above example) to preserve the contractive stability of the individual RNNs. To the best of our knowledge, this has not been done before in the machine learning and neuroscience literature.
> > >
> > > > “ensemble RNNs” , means ensembles of RNNs.
> > >
> > > Our approach differs from this approach in that we do not train many different models on the same task and then combine their predictions to get a "final" prediction. We only train one model (a single "RNN of RNNs") per task. However, our techniques are still relevant for ensemble approaches in the following sense: any weighted combination of outputs from a collection of contracting systems is itself contracting. Therefore, an ensemble of contracting RNNs is contracting as well.
> > >
> > > > "however, none of these models are guaranteed to be stable", how do we know other methods are not contractive stable?
> > >
> > > It is indeed possible that unconstrained models are globally contractive "by accident". However, this is extremely unlikely to happen by chance. This question was partially explored in Miller & Hardt, 2018 (https://arxiv.org/abs/1805.10369), where the authors found that unstable recurrent models were "stable in a data-dependent sense", meaning that they were locally (but not globally) stable. Global contractive stability allows us to prove many desirable properties of our networks (e.g robustness, combinability, etc).

---

### Official Review · Reviewer_tmsC · 2022-07-10

**Rating:** 4
**Confidence:** 4
**Soundness:** 3 good
**Presentation:** 2 fair
**Contribution:** 3 good

**Summary:**

The authors analyze a model of interconnected recurrent neural networks, trained to perform various tasks. They emphasize the importance of network stability to obtain good performance, and analyze it using  tools from contraction analysis. Specifically, they derive conditions under which a network of RNNs can remain stable. These, in turn, are used to parametrize the network for a training protocol that preserves stability during training.
This setup is used to test two different variants of a network of networks on the sequential MNIST and CIFAR tasks. Performance is higher than several similar models. Finally, the effect of hyper-parameters such as sparseness is numerically explored.


**Questions:**

1.	Benchmarks. I am not an expert in benchmarks of these form, but a quick search resulted in what seems like an RNN with a similar number of parameters and better performance (https://paperswithcode.com/paper/parallelizing-legendre-memory-unit-training/review/?hl=27784). Again - this is simply the first result I found.
2.	L289-290 “stability and modularity unexplored”. For instance: Aljadeff, Stern, Sharpee PRL 2015.
3.	L74-75. The linear coupling between nonlinear networks is not an intuitive choice from a neuroscience perspective. It is claimed that the nonlinear case will also be considered, but I didn’t find it.
4.	Equation 2 : is it x_i or x_j ?
5.	You show that modular networks learn better when initialized sparsely - this is an interesting result (perhaps related to the advantage of sparse connectivity in reservoir networks). Unfortunately, no intuition or analysis of why this is the case is provided.



**Limitations:**

Yes

**Strengths And Weaknesses:**

As recording from multiple brain areas simultaneously becomes more feasible in experiments, it is timely to study how networks of networks can collaborate to perform more complex and higher-level computations. This paper brings to the front the importance of making sure that certain properties of RNNs are retained when they are combined and assembled into a larger network.
The topic is thus timely and important.
Stability of trained networks has been recognized as important for performance (e.g., Sussillo & Abbott 2009), but mainly studied in the reservoir computing framework - where there is feedback to a single network. The current work extends this framework to the scenario of multiple networks.

The main weaknesses are the difficulty to understand the main objective of the paper, and a lack of reference to existing literature.
The motivation in the abstract is to better understand multi-area RNNs as models in neuroscience. It isn’t clear why benchmarks serve this purpose. On the other hand, if this is a proposal for a new ML architecture, it seems that performance isn’t better than state of the art.
Furthermore, the improved performance in the modular case isn’t accompanied by any analysis of the resulting networks. Do different modules handle different parts of the computation? Is the optimal number of modules related to the task? If so, how? These questions are much more relevant to the neuroscience motivation, but are not explored at all.

---

> ### Author Response · Authors · 2022-07-31
> **We thank the reviewer for raising several points which helped us clarify our contributions in the revised manuscript. We detail these clarifications in the comments below.**
>
> Part 1/2
>
> > The motivation in the abstract is to better understand multi-area RNNs as models in neuroscience. It isn’t clear why benchmarks serve this purpose. On the other hand, if this is a proposal for a new ML architecture, it seems that performance isn’t better than state of the art.
>
> Although our work is motivated primarily  by neuroscience, our approach is relevant for both neuroscience and machine learning. In neuroscience, our work is relevant in two distinct ways: 1) Because the Wilson-Cowan RNN is a time-tested model of neural circuit dynamics, the constraints we derive on the hidden weight matrices have direct neurophysiological interpretations. For example, the sparsity constraint of Theorem 1 provides another potential explanation for the highly sparse neural connections observed in cortex. 2) The sequential image recognition tasks we used are much more challenging than any neuroscience task one would typically use RNNs for (e.g delayed match to sample, random motion/color tasks, etc). Indeed, recent work has shown that in many cases cognitive neuroscience tasks can be solved without any recurrence between neurons at all (Khona, Chandra et al, arXiv, 2022). We reasoned that if our architecture could be trained to solve these challenging machine learning benchmarks, they could be easily trained to solve most neuroscience tasks involving recurrence. We plan to explore this more systematically in future work.
>
> In addition to being relevant for neuroscience, our work is also relevant for machine learning. Stability is a highly desirable property for recurrent architectures, for reasons related to safety and robustness (see e.g Revay & Manchester, 2020, Proceedings of Machine Learning Research). As explained in the paper (e.g L54-55), our claims of SOTA only apply to the model class of provably stable recurrent architectures. As the reviewer correctly points out, there are other models that perform better on the benchmarks we used. However, these models are not guaranteed to be stable.
>
> > Do different modules handle different parts of the computation? Is the optimal number of modules related to the task? If so, how?
>
> Unfortunately, due to space restrictions, we had to relegate many of these analyses to the appendix. For example, in A4.2.1 we explore how test accuracy after training varies as a function of the number of modules (while holding the total number of neurons fixed). In this analysis we do indeed find that there is an optimal number of modules, as evidenced by the upside-down U-shaped curve in Figure S1B. We intend to conduct a more thorough experimental analysis of this phenomenon (including the question of different tasks addressed above) in future work, where we do not have to “fight for space” with our novel theoretical results. We have edited the manuscript to better highlight the additional analyses in the appendix.

---

> > ### Author Response · Authors · 2022-07-31
> > **Part 2 of Our Initial Reply**
> >
> > Part 2/2
> >
> > > Benchmarks.
> >
> > As mentioned in the paper (e.g old manuscript L54-55) and in the reply above, we are only claiming SOTA within the class of provably stable recurrent architectures (denoted by the second column in Table 1). Additionally, the linked paper does not perform the sequential CIFAR10 task. This task is much more difficult than sequential MNIST, due to the longer sequence lengths, and consequently requires many more parameters.
> >
> > > L74-75...It is claimed that the nonlinear case will also be considered, but I didn’t find it.
> >
> > We have edited the manuscript to reflect that we only consider nonlinear hierarchical couplings (old manuscript, L101 and L544). It is an open and interesting question on how to include nonlinear negative feedback connections—something we are actively working on. We have explored the relationship between “classical” diagonal stability results and contraction in our networks, and found counterexamples (see section “What do the Jacobian Eigenvalues Tell Us?”) where diagonal stability does not imply contraction–this may partially underlie the difficulty of considering nonlinear negative feedback connections.
> >
> > We have also edited the manuscript to note that linear couplings are a valid approximation to nonlinear couplings around fixed points, in the same way that linear RNNs are valid approximations to nonlinear RNNs around fixed points.
> >
> > > You show that modular networks learn better when initialized sparsely...no intuition or analysis of why this is the case is provided.
> >
> > We agree: the sparse networks are indeed related to reservoir networks. For reservoir networks, typically only the outgoing weights are trained (e.g echo state networks). However, for our networks, we train both internal weights and outgoing weights. Nevertheless, the particular subset of internal weights we train correspond exactly to the outgoing weights of the individual RNNs. Thus, while the networks are related, there are important differences.
> >
> > It is also worth noting that in supplemental section A4.2.2 we perform additional experiments on sparsity. One positive contributing factor we found is that a sparse structure can allow for much higher magnitude weights to occur while still meeting overall stability constraints. The performance boost of sparsity is not as great when we restrict individual weight magnitudes to be as small as they would in the less sparse case.

---

> > > ### Comment · Reviewer_tmsC · 2022-08-09
> > > **Thanks for clarifications**
> > >
> > > Thank you for the clarifications, and I apologize for the late posting of my reply.
> > >
> > > After reading all reviews, all answers, and the relevant parts of the manuscript, I am still unconvinced about the contribution of the paper.
> > >
> > > If I understand correctly, there are two main claims:
> > >
> > > * Understanding modularity in the brain (or advancing this)
> > > * Machine learning: improvements in the realm of trainable, provably stable, architectures.
> > >
> > > I am not an expert on the second part, but it seems to me that there isn't a concensus in the field that non-provably-stable architectures are something to be avoided. On the contrary - I think there is a small body of work that emphasizes the importance of provably-stable. They could be correct! But the way it is presented is as if this is the only truth.
> > >
> > > Regarding neuroscience. I don't see this as a strong contribution. Yes - the current work tackles more challenging tasks than typical neuro papers. And the authors provide a framework to do this in a modular fashion. And it is shown that modularity can help. BUT - there is no attempt to understand why modularity helps, how it functions, do modules specialize in any way, etc. The U shaped graph - showing that there is some optimal hyperparameter - is nice and indicative of potential. But this is not understanding as I would expect in this type of work.

---

> > > > ### Author Response · Authors · 2022-08-09
> > > > **Thank you for further comments, we reply below:**
> > > >
> > > > >  I think there is a small body of work that emphasizes the importance of provably-stable. They could be correct! But the way it is presented is as if this is the only truth.
> > > >
> > > > With regards to machine learning, we are unfortunately not sure what is meant by "small body of work". [Here is a very recent review article from Manchester et al](https://arxiv.org/pdf/2110.00207.pdf) specifically focused on this area of research, which contains many references.
> > > >
> > > > We are also unaware that we presented "provably stable architectures" as the only truth--if we did so, this was certainly not intended. We would appreciate a specific pointer to where we made such a claim, so that we can correct it in the manuscript.
> > > >
> > > > > Regarding neuroscience. I don't see this as a strong contribution... there is no attempt to understand why modularity helps, how it functions, do modules specialize in any way, etc.
> > > >
> > > > Our main finding is, crucially, not that "modularity helps", nor is it "how modularity functions". Our main contribution is to show in what cases the stability of individual RNNs (a topic which is the central focus of hundreds of papers, see [Zhang et al](https://ieeexplore.ieee.org/stamp/stamp.jsp?tp=&arnumber=6814892) for a comprehensive overview) is preserved when embedding them in a large, recursive, feedback-heavy structure like the brain. To our understanding, this question has not been asked (or addressed) in the neuroscience literature. We believe it is important and timely to ask it, because experimental neuroscience is quickly moving into multi-area recordings being the standard operating procedure.
> > > >
> > > > We stress that our empirical findings are there for two reasons: 1) to show that stability does not necessarily preclude good performance on challenging tasks and 2) to show that our stability-preserving combination conditions are parameterized in a way that is amenable to gradient-based optimization.
> > > >
> > > > Thank you again for taking the time to respond to us. We hope that the above comments clarified some important points of our paper.

---

### Official Review · Reviewer_vKbB · 2022-07-13

**Rating:** 8
**Confidence:** 2
**Soundness:** 4 excellent
**Presentation:** 4 excellent
**Contribution:** 4 excellent

**Summary:**

The authors consider the behaviour of networks of artificial recurrent neural networks (RNNs), specifically the stability of RNNs of RNNs. This is achieved using nonlinear control theory, specifically to consider the contraction properties of the networks (where a contracting system is one in which it's long-term behaviour is exponentially independent of initial conditions).



**Questions:**

Is it possible to discern the role of different sub-networks in forming learned representations? What is the effect of ablating sub-networks during training?

How well does the results on stability and contractivity apply to other network architecture? Could a similar theoretical analysis be applied to other recurrent models e.g. neural cellula automata?

**Limitations:**

It wasn't immediately clear to me how well their results generalise to other network architectures. I also note this as a question to the authors above.

**Strengths And Weaknesses:**

The authors identify that understanding networks of neural networks is significant both for using all available neural data in neuroscience as well as for new deep learning architecture frameworks. They use nonlinear control theory to analyze the contraction of networks of RNNs. I believe that this, in itself is novel and is further improved on by them deriving several original results. They also review several recent results in machine learning theory, such as the sufficient conditions for stability of nonlinear systems and give clarification to their applicability. The authors implement a form of RNNs of RNNs which are expected to satisfy appropriate contractivity, given their theoretical exposition. They find strong performance when compared to other baselines.

Overall, I did not find signficant weaknesses. The manuscript was well structured, on an original topic, and the authors were thorough in their investigation.

---

> ### Author Response · Authors · 2022-07-31
> **We thank the reviewer for their positive assessment of our paper. We answer their two questions below:**
>
> > Is it possible to discern the role of different sub-networks in forming learned representations? What is the effect of ablating sub-networks during training?
>
> In our experiments, we found that responses were distributed across all sub-networks, without any obvious pattern. However, this may be a consequence of the benchmark tasks we used. An interesting future direction would be to try a “multi-modal” task, and to see if different sensory modalities are naturally funneled into different subnetworks.
>
> While we did not perform explicit ablation experiments, our experiments showing how test accuracy varies as a function of the number of sub-networks (Figure S1B) suggests that for large networks, ablating subnetworks early in training may have a beneficial role. This follows from the upside down U-shape of the curve–if we are too far to the right on the x-axis (i.e too many subnetworks) test accuracy starts to drop. Ablation would correspond to moving left on the x-axis, thereby potentially increasing the test accuracy.
>
> > How well does the results on stability and contractivity apply to other network architecture? Could a similar theoretical analysis be applied to other recurrent models e.g. neural cellula automata?
>
> The results apply equally well to any other recurrent architecture, so long as one can show contractivity of the individual models before combining them. For example, (Miller & Hardt, 2019) provides contractivity conditions for LSTMs in the identity metric—therefore these models can be used as a drop-in replacement for the particular RNN models we have used here, which were chosen on the basis of neuroscientific relevance.

---

### Comment · Area_Chair_4eg6 · 2022-08-07
**reviewer-author discussion**

Dear reviewers,

The authors have posted up their rebuttals.  If you have not done so, please engage with the authors to discuss your feedback or any further concerns.

Thanks,

AC

---

### Meta-Review · Area_Chair_4eg6 · 2022-08-25

**Recommendation:** Accept
**Confidence:** Certain

**Metareview:**

In this paper the authors propose a so-called "RNN of RNNs"  architecture where an assembly of contractively stable RNNs can still work collaboratively under contractive stability.  The authors draw the motivation from neuroscience where multiple brain areas can work simultaneously for complex behaviors. Conditions are investigated under which multiple interacting RNNs with feedback connections will preserve contractive stability.  Experiments are carried out on seqMNIST, permuted seqMNIST and seqCIFAR datasets. It is shown that the proposed network of RNNs can give decent results. Notably, SOTA performance is claimed within the class of provably stable recurrent architectures.  All reviewers consider the work theoretically solid and timely to the community.  Authors' rebuttal clears most of the concerns.  The paper can be accepted.  One lingering concern which, however, still stands after the rebuttal and discussion is its connection and impact to neuroscience where the motivation is drawn.  Following the authors' narrative, the work would be more convincing if the proposed "RNN of RNNs" can show effectiveness directly on neuroscience datasets.  The authors should revise the paper accordingly to reflect that.

**Award:**

No

---

### Decision · Program_Chairs · 2022-09-14

Accept